# PKA-RIIβ autophosphorylation modulates PKA activity and seizure phenotypes in mice

Jingliang Zhang [1,8], Chenyu Zhang[2,8], Xiaoling Chen[1], Bingwei Wang[2], Weining Ma[3], Yang Yang[4], Ruimao Zheng[2,5,6,7,9 ✉] & Zhuo Huang[1,6,7,9 ✉]

Temporal lobe epilepsy (TLE) is one of the most common and intractable neurological disorders in adults. Dysfunctional PKA signaling is causally linked to the TLE. However, the mechanism underlying PKA involves in epileptogenesis is still poorly understood. In the present study, we found the autophosphorylation level at serine 114 site (serine 112 site in mice) of PKA-RIIβ subunit was robustly decreased in the epileptic foci obtained from both surgical specimens of TLE patients and seizure model mice. The p-RIIβ level was negatively correlated with the activities of PKA. Notably, by using a P-site mutant that cannot be autophosphorylated and thus results in the released catalytic subunit to exert persistent phosphorylation, an increase in PKA activities through transduction with AAV-RIIβ-S112A in hippocampal DG granule cells decreased mIPSC frequency but not mEPSC, enhanced neuronal intrinsic excitability and seizure susceptibility. In contrast, a reduction of PKA activities by *RIIβ* knockout led to an increased mIPSC frequency, a reduction in neuronal excitability, and mice less prone to experimental seizure onset. Collectively, our data demonstrated that the autophosphorylation of RIIβ subunit plays a critical role in controlling neuronal and network excitabilities by regulating the activities of PKA, providing a potential therapeutic target for TLE.

[1] State Key Laboratory of Natural and Biomimetic Drugs, Department of Molecular and Cellular Pharmacology, School of Pharmaceutical Sciences, Peking University Health Science Center, Beijing, China. [2] Department of Anatomy, Histology and Embryology, School of Basic Medical Sciences, Peking University Health Science Center, Beijing, China. [3] Department of Neurology, Shengjing Hospital Affiliated to China Medical University, Shenyang, China. [4] Department of Medicinal Chemistry and Molecular Pharmacology, Purdue University College of Pharmacy, West Lafayette, IN, USA. [5] Neuroscience Research Institute, Peking University, Beijing, China. [6] Key Laboratory for Neuroscience, Ministry of Education, Beijing, China. [7] Key Laboratory for Neuroscience of National Health Commission, Beijing, China. [8] These authors contributed equally: Jingliang Zhang, Chenyu Zhang. [9] These authors jointly supervised this work: Ruimao Zheng, Zhuo Huang. ✉email: rmzheng@pku.edu.cn; huangz@hsc.pku.edu.cn

Temporal lobe epilepsy (TLE) is one of the most common forms of drug-resistant epilepsy in adults. There is strong evidence that the hippocampus and entorhinal cortex play pivotal roles in the induction and maintenance of TLE[1,2]. However, the mechanism underlying the hyperexcitability phenotype of the hippocampus is still elusive.

PKA, as a key serine/threonine kinase and central element of cellular signal transduction, is extensively expressed across multiple brain regions. The high level of PKA expression is detected in the hippocampus and neocortex[3–5]. Importantly, PKA function is essentially involved in the regulation of neuronal and network activities[6,7]. Clinically and biologically, numerous reports unveiled that the increased PKA activities are causally linked to the pathological development of epileptic seizures[8–11]. For instance, emerging evidence points to an important role for PKA in the genesis of epilepsy[12–17], and activation of PKA can maintain an epileptic activity or increase seizure susceptibility in vivo[8,18]. Besides, genome-wide association studies (GWASs) revealed that the PKA/CREB signaling pathway is genetically associated with focal epilepsy from the human hippocampus samples[19]. Taken together, these studies implicate that PKA plays a critical role in the pathogenesis of epilepsy. However, the mechanism underlying the participation of PKA in epileptogenesis is not well understood and whether the PKA signal pathway could become an effective target in epilepsy treatment needs to be urgently investigated.

PKA holoenzyme is a heterotetramer that consists of a dimer of two regulatory (R) subunits with each binding a catalytic (C) subunit[20]. The binding of cAMP to the regulatory subunits causes its conformational alteration, leading to the release of active C subunits. The liberated C subunits phosphorylate protein substrates at serine, or threonine residues, resulting in changes in cell function, such as neuronal excitabilities[21]. In mammals, four R subunit genes (encoding RIα, RIβ, RIIα, and RIIβ) and two C subunit genes (encoding Cα and Cβ) have been identified. RIα, RIβ, and RIIβ are highly expressed in the brain while RIIα is almost exclusively in the medial habenular nuclei[3,22–25]. In neurons, RIIβ is anchored to adenosine kinase anchoring proteins (AKAPs) in the cell membrane, which confers a stronger ability to RIIβ subunit to regulate the phosphorylation level of ion channels and receptors of neurotransmitters[22,25–28], while RIα and RIβ are localized in cytoplasm[26]. Strikingly, RIIβ can be autophosphorylated at a serine, namely Ser112 in mice or Ser114 in humans, to dynamically downregulate PKA catalytic activity in the neurons[29–33]. Collectively, these observations imply that RIIβ may serve as a key factor in the regulation of neural excitabilities, including intrinsic excitabilities and network activities of neurons. Therefore, investigating the function of the PKA-RIIβ subunit during the epileptic seizures may provide a novel insight into the regulation of PKA signaling in epileptogenesis.

In the present study, we have found the autophosphorylation levels at both the Ser114 site in humans and the Ser112 site in mice of PKA-RIIβ subunit were decreased in epileptic foci obtained from both surgical resections of TLE patients and mice subject to experimental temporal lobe seizures. Further, we observed a reverse correlation between the RIIβ autophosphorylation level and seizure susceptibility. Mechanistically, the reduced autophosphorylation level of RIIβ in vivo could increase neuronal intrinsic excitabilities and decrease the miniature IPSCs frequency in hippocampal DG granule cells. Consistently, reduced PKA activities by PKA-RIIβ subunit knockout led to an increased miniature IPSCs frequency in DG granule cells and a reduction in neuronal intrinsic excitability, and mice less prone to seizure onset. Together, our data indicate that the PKA signal is a critical factor that involves seizure genesis by modulating neuronal and network excitabilities, providing a potential therapeutic target for TLE.

## Results

### Enhanced PKA activities and reduced RIIβ autophosphorylation level were found in epileptic foci of TLE patients and experimental seizure model mice.

Previous findings showed that upregulated cAMP/PKA remarkably enhanced seizure susceptibility[15,34,35], and the activity of cAMP/PKA is dynamically controlled by a process of autophosphorylation on RIIβ[36]. To confirm the role of PKA activities in epilepsy, we used an antibody that can recognize the phosphorylated substrates of PKA and found robust enhancement of PKA activities in epileptic foci obtained from surgically removed human anterior temporal lobe and hippocampal tissues (Fig. 1a, b, Supplementary Figs. 5a–d, 1a and 9c, i, o, and Supplementary Table 1). To investigate whether the autophosphorylation of RIIβ could play a role in regulating the PKA activities, we examined the expression level of phosphorylated RIIβ (p-RIIβ) and total RIIβ in epileptic foci of TLE patients. Compared to adjacent normal tissues, 13 out of 14 matched pairs of TLE samples have a decreased RIIβ autophosphorylation level at the Ser114 site (Fig. 1c, d and Supplementary Figs. 6a, c, 1a, and 9a, g, m), whereas the total RIIβ expression level was unaltered (Fig. 1c, e and Supplementary Figs. 6b, d, 1a, and 9b, h, n). The cAMP-response element-binding protein (CREB) is a key nuclear transcription factor in downstream of the PKA signal pathway. Generally, the level of phosphorylated CREB is positively linked to PKA activities. Therefore, to further confirm the role of PKA activities in the brain tissues of epilepsy patients, we examined the phosphorylation level of CREB at the Ser133 site which is catalyzed by PKA catalytic subunit[37–40]. In the hippocampi of these TLE patients, we observed the CREB phosphorylation level at the Ser133 was remarkably augmented, and the total CREB was unchanged, which indicated that the surged PKA catalytic activities, but not the total protein levels in epileptic foci of TLE might play a critical role during the epileptogenesis (Fig. 1f, g and Supplementary Figs. 6e, f, h, I, 1a, and 9d, e, j, k, p, q). Taken together, these observations suggest that the raised PKA activities are positively associated with epileptogenesis, and the phosphorylation level at the Ser114 site of the RIIβ subunit may be closely involved in regulating the PKA activities.

To further investigate the role of PKA in seizure genesis, we used a well-documented kainate model, as previously described[41–43]. In this model, a single episode of Class V seizures (as defined by the Racine scale[44]) or status epilepticus (SE) is induced in rodents by administering kainic acid (KA) and then terminated 1 h later with an anticonvulsant such as sodium pentobarbital (see Experimental procedures). After a delay of a few weeks, known as the latent period (during which animals appear to be normal), spontaneous overt electrophysiological and/or behavioral seizures occur (defined as the onset of chronic seizures)[41–43,45–47]. This model is widely used because many of the clinical and pathological features of the human disorder (including the latent period) can be reproduced[41–43].

Two-weeks after status epilepticus, we found that the RIIβ autophosphorylation level at the Ser112 site was significantly decreased in DG, CA1, CA3 regions of the hippocampus and entorhinal cortex from KA-induced experimental temporal lobe seizure model mice, whereas the total RIIβ protein expression was unchanged (Fig. 1h–j and Supplementary Fig. 7a–f). Our analysis revealed that the phosphorylation level at Ser133 of CREB was significantly augmented in the experimental seizure model mice compared with the control mice, whereas the total CREB protein level was not changed (Fig. 1h, k, l and Supplementary Fig. 7g–i). The PKA activities in the experimental seizure model were detected with the antibody which recognizes the whole phosphorylated substrates of PKA. As expected, when compared to temporal lobe tissues from control mice, we found the level of

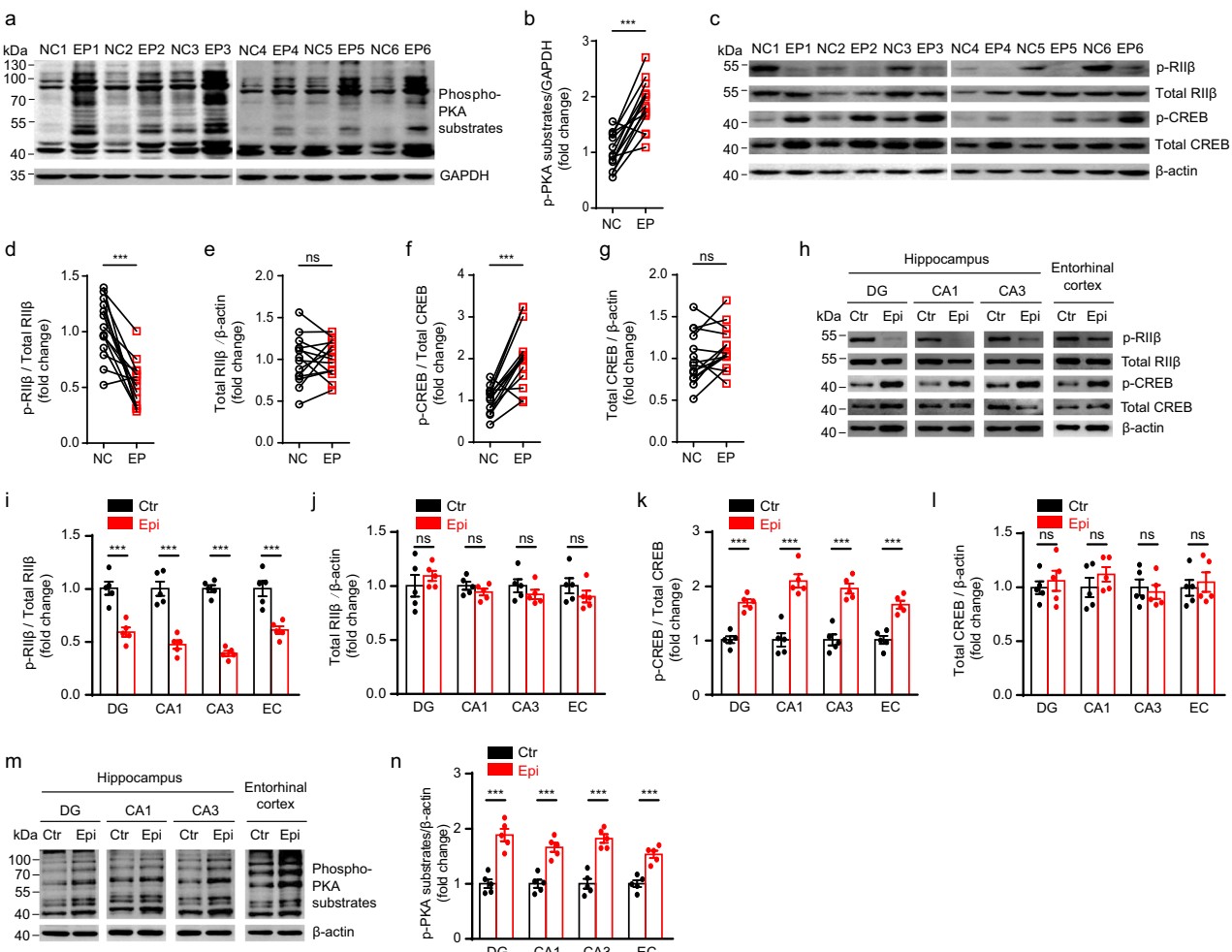

**Fig. 1 Decreased autophosphorylation of the RIIβ subunit of PKA and increased phosphorylation of CREB and PKA substrates in epileptic foci of TLE patients and experimental seizure model mice. a** Western blot analysis of p-PKA substrates. NC normal control using adjacent normal tissues from a designated epilepsy patient, EP epileptic focus from a designated epilepsy patient. If the EP was the hippocampus, then the NC was the entorhinal cortex embedded in the anterior temporal lobe; if the EP was the entorhinal cortex in the anterior temporal lobe, then the NC was the hippocampus. **b** Quantification of the Western blot results in **a** and Supplementary Fig. 1a, $n = 14$ patients, ***$P < 0.001$, paired two-tailed Student's $t$-test. **c** Western blot analysis of p-RIIβ, total RIIβ, p-CREB, and total CREB in the epileptic foci and adjacent normal tissues obtained from the surgically removed human anterior temporal lobe and hippocampus. **d–g** Quantification of the Western blot results in **c** and Supplementary Fig. 1, $n = 14$ patients, ns no significance, $P > 0.05$; ***$P < 0.001$, paired two-tailed Student's $t$ test. **h** Western blot analysis of p-RIIβ, total RIIβ, p-CREB, and total CREB in the DG, CA1, CA3 area, and entorhinal cortex from mice in control (Ctr) or experimental seizure model (Epi) group. **i–l** Quantification of the Western blot results in H ($n = 5$ mice for each group). ns no significance, $P > 0.05$; ***$P < 0.001$, unpaired two-tailed Student's $t$ test. **m** Western blot analysis of p-PKA substrates in the DG, CA1, CA3 area, and entorhinal cortex from mice in control or experimental seizure model group. **n** Quantification of the Western blot results in **m** ($n = 5$ mice for each group). ***$P < 0.001$, unpaired two-tailed Student's $t$ test. Data were represented as mean ± SEM.

phospho-PKA substrates was remarkably increased in DG, CA1, CA3, and EC regions from mice subject to experimental temporal lobe seizures (Fig. 1m, n and Supplementary Fig. 5e–h). These data indicate that the reduction of RIIβ autophosphorylation at site Ser112 causes an elevation of PKA activities.

**Reduced autophosphorylation of RIIβ in vivo increased PKA activities and seizure susceptibility in mice.** To test our hypothesis that the elevation of PKA activities due to the reduction of RIIβ autophosphorylation at the site of Ser112, could facilitate the development of acute experimental seizures, we designed a model, in which the RIIβ autophosphorylation was consistently suppressed by a non-phosphorylatable mutation. As in the hippocampus, the RIIβ shows a higher expression level in the dentate gyrus than that in the CA1-3 cell layers[3], using an adeno-associated viral vector (AAV) delivered by stereotaxic

injection, we mutated the serine 112 to alanine in the RIIβ subunit (AAV-RIIβ-S112A, a P-site mutant that cannot be phosphorylated) with a broadly expressed promoter CMV, to impede the autophosphorylation in the DG area of the murine hippocampus. We observed that the AAV-RIIβ-S112A was robustly expressed in almost all cells in the DG area in the third week after the injection (Fig. 2a and Supplementary Fig. 2a, c), and the level of autophosphorylation of RIIβ in the AAV-RIIβ-S112A positive neurons was significantly reduced when compared to the neurons transfected with control AAV vector (Fig. 2b, c and Supplementary Fig. 2b, d). Consistently, the suppression of RIIβ autophosphorylation caused a significant increase in the phosphorylation level of CREB at Ser133 and total PKA phosphosubstrates, compared with the control groups (Fig. 2b, e, g and Supplementary Fig. 2d, e). No significant differences were observed in total RIIβ or total CREB protein levels (Fig. 2b, d, f and Supplementary Fig. 2d).

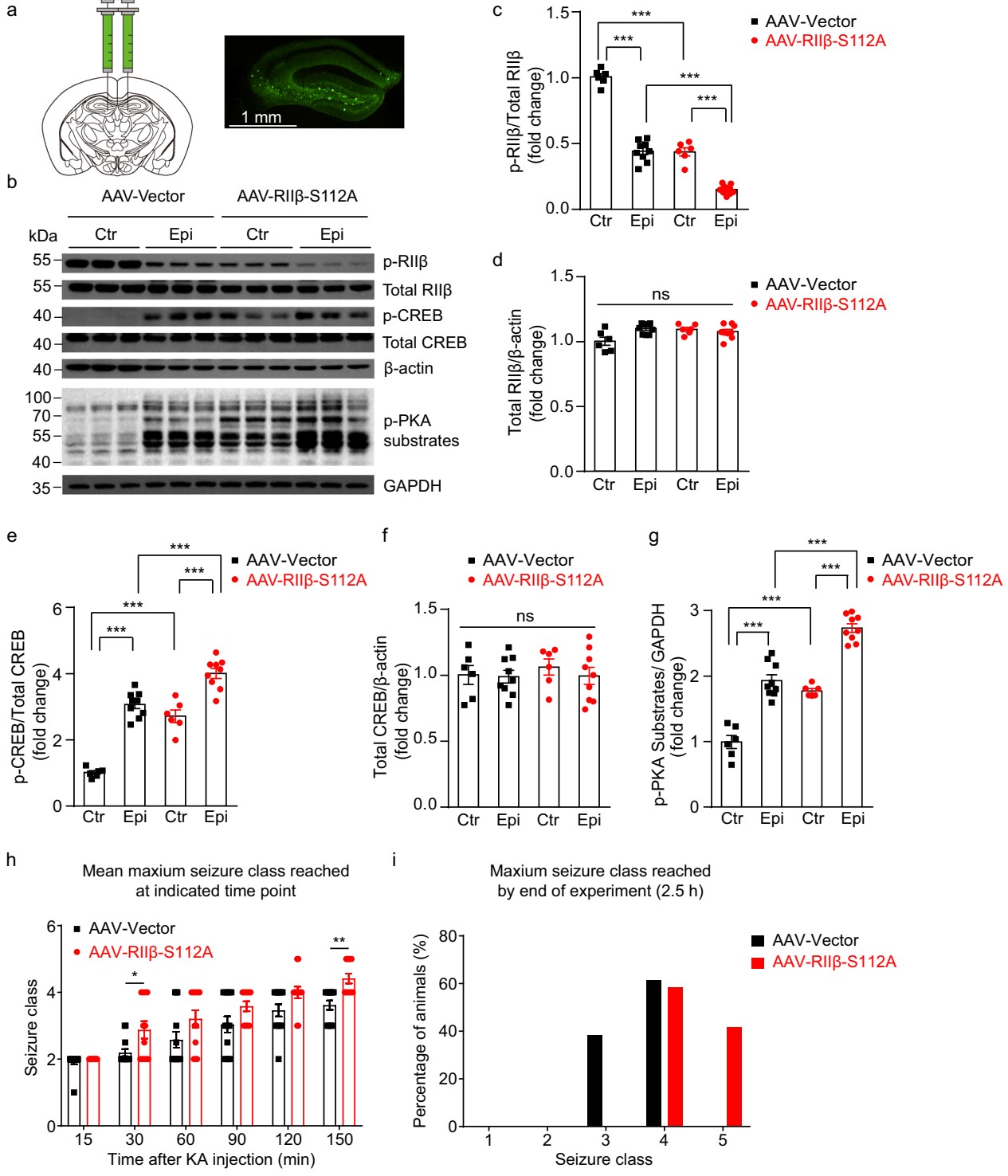

**Fig. 2 Downregulation of RIIβ autophosphorylation in vivo led to increased phosphorylation of CREB and PKA substrates, as well as increased seizure susceptibility in mice. a** Schematic diagram of injection and expression of adenosine associated virus in the hippocampal DG area. scale bar, 1 mm. **b** The Western blot analysis showed p-RIIβ, total RIIβ, p-CREB, total CREB, and p-PKA substrates protein levels in the hippocampal DG area from mice in AAV-Vector or AAV-RIIβ-S112A group. Ctr control, Epi experimental seizure model. **c–g** Quantification of the Western blot results in **b**. $n = 6$ mice for each group of Ctr and $n = 9$ mice for each group of Epi, ns no significance, $P > 0.05$; ***$P < 0.001$, one-way ANOVA with Bonferroni's multiple-comparison test. **h** Graph depicting limbic seizure progression, illustrated as mean maximum seizure class reached by 15, 30, 60, 90, 120, or 150 min after kainic acid administration (30 mg/kg, i.p.) in AAV-Vector transfected ($n = 13$) and AAV-RIIβ-S112A transfected ($n = 12$) mice. $P = 0.0379$ at 30 min, $P = 0.074$ at 120 min, $P = 0.0025$ at 150 min, $P < 0.05$; **$P < 0.01$, for each time bin, data were analyzed by unpaired two-tailed non-parametric Mann–Whitney U-test. **i** Incidence of maximum seizure class reached during the course of the experiment. Data were represented as mean ± SEM.

Since RIIβ autophosphorylation may play an important role in the progression of seizures through regulating PKA activities, to further assess the pathophysiological consequences of reduced autophosphorylation of RIIβ in vivo, we used the KA-induced seizure model to assess the seizure susceptibility[48]. Intraperitoneal administration of KA 30 mg/kg elicited status epilepticus (SE) (≥Class 4 seizures, see Kainic acid-induced temporal lobe seizures section in Experimental procedures) in a majority of mice (Fig. 2h, i). AAV-RIIβ-S112A transfected mice exhibited a more serious seizure progression, measured by maximum seizure class reached in each successive bin of 30 min compared with AAV-Vector transfected mice (Fig. 2h). In the AAV-Vector transfected group, 8/13 of mice developed into generalized seizures, whereas 12/12 of mice in the AAV-RIIβ-S112A transfected group progressed into generalized seizures (≥Class 4 seizures: $P < 0.05$, chi-square). Moreover, the maximum seizure severity was considerably higher in AAV-RIIβ-S112A transfected mice, with more of them developing into tonic hindlimb extension and death (Class 5) compared with AAV-Vector transfected mice ($P < 0.01$, chi-square) (Fig. 2i). These results indicated that the downregulation of RIIβ autophosphorylation in vivo increased the phosphorylation levels of CREB and PKA substrates, which may contribute to the enhancement of the seizure onset.

As epileptic seizures are closely related to the neuronal activities[49], to exclude the effect of RIIβ expression in non-neuronal cells, we have investigated the expression level of RIIβ in GFAP-positive cells using immunostaining. Our results showed that the expression levels of RIIβ in GFAP-positive cells were very low, suggesting that RIIβ subunits are rarely localized to astrocytes (Supplementary Fig. 1b). Thus, the neurons seem to be the main workplace for the RIIβ-PKA activities, considering that RIIβ is abundantly expressed in neurons[50].

**RIIβ null mice show decreased PKA activities and reduction in seizure susceptibility.** It was previously reported that p-CREB expression was decreased in the hypothalamus in RIIβ null mice[51]. To determine whether the knockout of RIIβ subunit could also induce a reduction in the p-CREB level and PKA activities in the temporal lobe, and thus acts as an anti-seizure factor in mice, we assessed CREB phosphorylation level at Ser133 in DG, CA1, CA3, and EC from $RIIβ^{-/-}$ mice. Our Western blot analysis showed that the levels of p-CREB and p-PKA substrates were markedly decreased in the hippocampal regions as well as the entorhinal cortex in $RIIβ^{-/-}$ mice (Fig. 3a, b, d and Supplementary Fig. 8), while the total CREB protein level was unchanged (Fig. 3c). Importantly, these results demonstrated that disruption of $RIIβ$ gene expression could recapitulate the downregulation of PKA activities in the temporal lobe region, which might contribute to understanding the mechanism by which the downregulation of PKA activity resists epilepsy. Moreover, we further observed that RIIβ null mice exhibited significantly slower seizure progression, compared with WT mice (Fig. 3e). In the WT group, 7/8 of mice developed into generalized seizures, whereas 2/7 of mice in the $RIIβ^{-/-}$ group progressed into generalized seizures (≥Class 4 seizures: $P < 0.05$, chi-square). Notably, the maximum seizure severity was obviously lower in $RIIβ^{-/-}$ mice, with less of them developing into tonic hindlimb extension and death (Class 5) compared with WT mice ($P < 0.01$, chi-square) (Fig. 3f). These results suggested that PKA activities in epileptic seizure foci may be causally linked to the occurrence of acute seizures after KA administration in mice. The pathologically increased PKA activity in the neurons of the primary foci, such as the hippocampus or entorhinal cortex may be implicated in the genesis of seizures.

**Reduced autophosphorylation level at the RIIβ Ser112 site decreased mIPSCs in hippocampal DG granule cells.** To further study the effect of autophosphorylation of RIIβ on synaptic transmissions, we examined the miniature release of GABA and glutamate in acute DG slices using whole-cell voltage-clamp recordings. Hippocampal DG granule cells were identified by their typical location and complex dendritic trees with a large input resistance of $188.2 ± 8.6$ ($n = 17$), which is consistent with previous investigations (Supplementary Table 8)[52]. Miniature inhibitory postsynaptic current (mIPSC) was recorded by blocking excitatory synaptic transmissions with AP5 (NMDA receptor antagonist, 50 μM) and CNQX (AMPA/kainate glutamate receptor antagonist, 10 μM); as well as the miniature excitatory postsynaptic current (mEPSC), which was recorded by blocking inhibitory synaptic transmissions with bicuculline ($GABA_A$ receptor antagonist, 10 μM) and CGP55845 ($GABA_B$ receptor antagonist, 2 μM). In the presence of 500 nM tetrodotoxin (TTX), we found the mIPSC frequency recorded from AAV-RIIβ-S112A transfected neurons ($2.69 ± 0.20$, $n = 17$) was significantly decreased compared with AAV-Vector transduced neurons ($1.72 ± 0.16$, $n = 18$, $P < 0.001$) (Fig. 4a–c) without any detectable changes in other mIPSC parameters (Supplementary Table 2), while the mEPSC obtained from AAV-RIIβ-S112A transfected DG granule cells were not significantly affected (Fig. 4d–f and Supplementary Table 3).

On the contrary, the mIPSC frequency recorded from $RIIβ^{-/-}$ DG granule cells ($3.04 ± 0.27$, $n = 16$, $P < 0.05$) was significantly increased when compared with wild-type neurons ($2.08 ± 0.23$, $n = 15$), while the mEPSC was not significantly changed (Fig. 4g, l and Supp Tables 4 and 5). These results indicated that the downregulation of the autophosphorylation level of RIIβ dampens the inhibitory afferent onto hippocampal DG granule cells.

**Reduced autophosphorylation level of RIIβ in vivo increased neuronal intrinsic excitabilities of DG granule cells in mice.** Membrane-residing PKA substrates play an important role in regulating the activities of ion channels localized to different neuronal compartments, thus modulate the neuronal intrinsic excitabilities[53–57], such as the summation of postsynaptic signals along the neuronal dendrites, signal propagation from dendrites to axonal initial segments, and generation of action potentials[58]. Therefore, we next determined whether the reduced autophosphorylation of RIIβ affects intrinsic neuronal excitabilities. We used whole-cell current-clamp recordings in acute brain slices in the presence of antagonists of glutamate and GABA receptors. The average resting membrane potential (RMP) of AAV-RIIβ-S112A transduced neurons was much more depolarized ($-70.3 ± 1.1$ mV, $n = 12$) than that of AAV-Vector transduced neurons ($-76.3 ± 0.6$, $n = 12$, $P < 0.001$) (Fig. 5e). DG granule cells transduced with AAV-vector fired typical action potentials (APs) at normal RMP in response to a series of 400-ms current steps from $-200$ pA to $+300$ pA in the increment of 50 pA. When neurons were held at their normal RMPs, AAV-RIIβ-S112A transduced neurons fired more action potentials (APs) compared with the AAV-vector transduced neurons (Fig. 5a, b), while the differences in neuronal excitabilities were abolished when the neurons were held at a fixed membrane potential of $-80$ mV (Fig. 5c, d), suggesting that the increase in neuronal excitabilities mainly results from the RMP depolarization caused by the decreased autophosphorylation of RIIβ (Fig. 5e–g). Furthermore, we performed whole-cell current-clamp recordings with a minimum positive current injection to induce a single intact AP (Fig. 5h). This method allowed the measurement of neuronal AP properties without inactivating voltage-gated ion channels. The

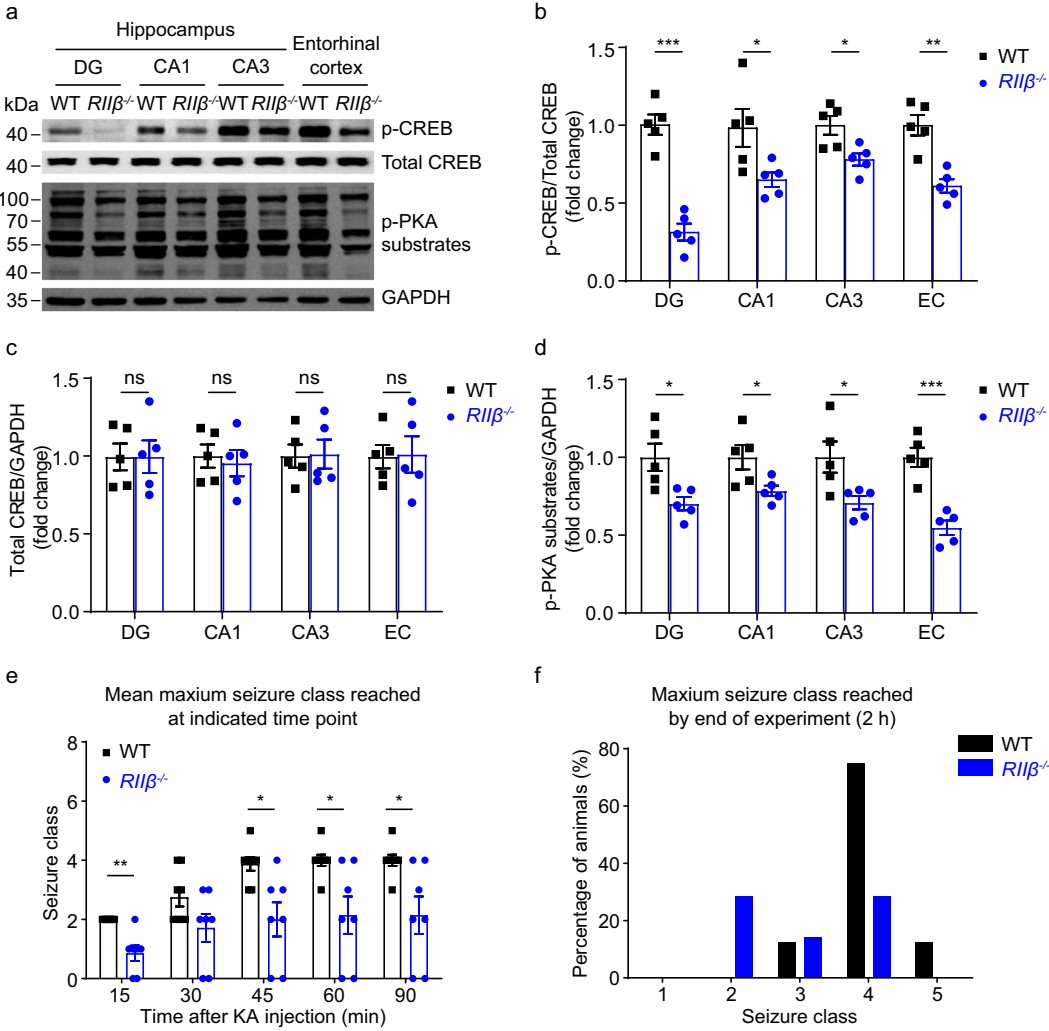

**Fig. 3 Decreased phosphorylation of CREB and PKA substrates in *RIIβ⁻ᐟ⁻* mice, and *RIIβ* null exhibited anticonvulsant activity in KA-induced mouse experimental seizure model. a** Western blot analysis showed p-CREB, total CREB, and p-PKA substrates protein levels in the DG, CA1, CA3 area, and entorhinal cortex (EC) from WT and *RIIβ⁻ᐟ⁻* mice. **b–d** Quantification of the Western blot results in **a** $n = 5$ mice for each group, *$P < 0.05$; **$P < 0.01$; ***$P < 0.001$, unpaired two-tailed Student's $t$ test. **e** Graph depicting limbic seizure progression, illustrated as mean maximum seizure class reached by 15, 30, 45, 60, or 90 min after kainic acid administration (30 mg/kg, i.p.) in WT ($n = 8$) and *RIIβ⁻ᐟ⁻* ($n = 7$) mice. $P = 0.0014$ at 15 min, $P = 0.1636$ at 30 min, $P = 0.0107$ at 45 min, $P = 0.0145$ at 60 min, $P = 0.0145$ at 90 min, ns no significance, $P > 0.05$; *$P < 0.05$; **$P < 0.01$ for each time bin, data were analyzed by unpaired two-tailed non-parametric Mann–Whitney $U$-test. **f** Incidence of maximum seizure Class reached during the course of the experiment. Data were represented as mean ± SEM.

results showed that transduction of AAV-RIIβ-S112A lowered the AP amplitude values at their normal RMPs (Fig. 5h, i, l) while the amplitudes were restored to the same level when both groups of neurons were held at a membrane potential of −80 mV (Supplementary Fig. 3e). Whereas other intrinsic membrane properties were not significantly affected when held either at the normal RMP or at a fixed potential of −80 mV (Fig. 5j, k, m, Supplementary Fig. 3a–f, and Supplementary Tables 6 and 7).

To further examine the effect of the reduced enzyme activities of PKA enzyme by *RIIβ* null on neuronal excitabilities in vivo, we subsequently tested intrinsic properties of DG granule cells from *RIIβ* null mice and their wild-type (WT) littermates. We found that when neurons were held at their normal RMPs, *RIIβ⁻ᐟ⁻* neurons fired fewer action potentials (APs) compared to WT neurons (Fig. 6a, b). This could be due to hyperpolarized RMPs (Fig. 6e), smaller input resistances (Fig. 6f), and hyperpolarized fAHP values of *RIIβ⁻ᐟ⁻* neurons (Fig. 6m), although other intrinsic membrane properties were not significantly altered in these cells (Fig. 6k, l and Supplementary Table 8). However, the

reduced neuronal intrinsic excitability in *RIIβ⁻ᐟ⁻* neurons could not be completely reversed by holding these neurons at the potential of −80 mV (Fig. 6c, d, g, Supplementary Fig. 4a–f and Supplementary Table 9). Together, the above data indicated that the reduced autophosphorylation of RIIβ increased neuronal intrinsic excitabilities of DG granule cells in vivo, which may be mainly through regulating neuronal RMPs.

## Discussion
In this study, we reported a reduced autophosphorylation level of human RIIβ at Ser114 site in epileptic foci of TLE patients and murine RIIβ at Ser112 site in experimental seizure model mice, which is concurrent with the activation of PKA (Fig. 1). Decreasing autophosphorylation level of mouse RIIβ at Ser112 site by transduction with AAV-RIIβ-S112A in hippocampal DG granule cells led to an increased level of PKA activities and a higher seizure susceptibility, whereas knockout of *RIIβ* exhibited decreased PKA activities and less prone to seizure onset (Figs. 2 and 3). Mechanistically, we show a reduction in

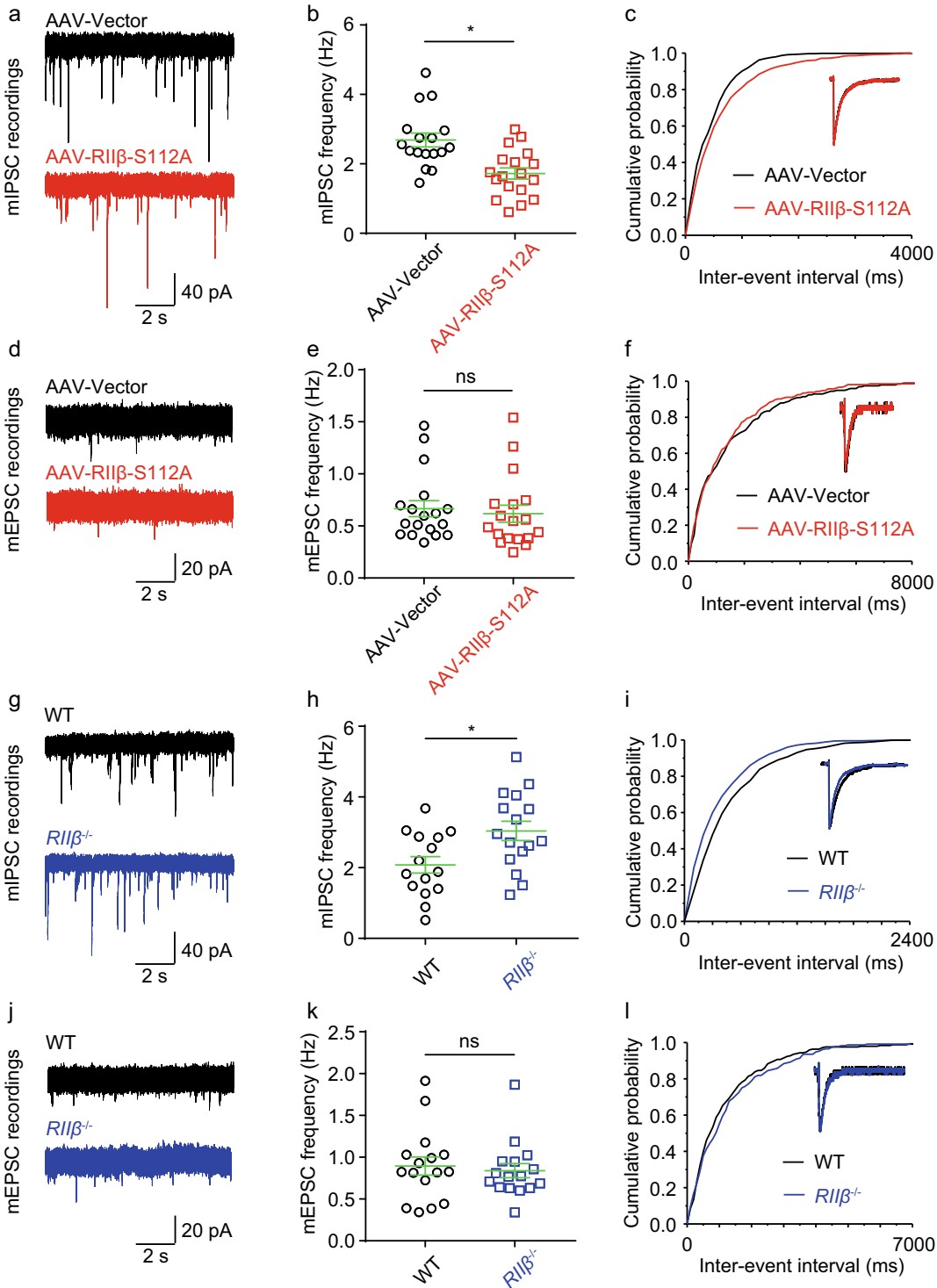

autophosphorylation level of RIIβ in vivo increased neuronal excitabilities (Figs. 5 and 6), and decreased mIPSCs frequency in hippocampal DG granule cells (Fig. 4), thus facilitating the development of mouse experimental seizures.

RII inhibitor sites have a serine at their phosphorylation site (P site) and are both substrates and inhibitors[33], phosphorylation of the P-site Ser in RII slows the rate of association with C subunit, and formation of holoenzyme in cells is influenced substantially depending on whether the P-site residue is a substrate or a pseudosubstrate[33]. Thus, the signal transduction activity of

cAMP-PKA is dynamically controlled by a process of autophosphorylation on the RIIβ subunit, which leads to a robust downregulation of PKA activities and thereby contributes to the suppression of neuronal excitabilities.

Our proposed mechanism underlying the increased PKA activities in TLE patients and KA-induced experimental seizure model mice (Fig. 7) is that in physiological conditions, the PKA-C subunits can autophosphorylate RIIβ subunits using MgATP, and the phosphorylated RIIβ can turn back into unphosphorylated RIIβ using protein phosphatases (PPs). The PKA holoenzyme

**Fig. 4 Downregulation of autophosphorylation of RIIβ decreased inhibitory afferent, and _RIIβ_ null increased inhibitory afferent of DG granule cells without changing excitatory afferent in mice. a** Representative raw recording of miniature inhibitory postsynaptic currents (mIPSCs) in DG neurons transfected with AAV-Vector (upper trace, black) and AAV-RIIβ-S112A (lower trace, red). **b** Comparison of the average values ± SEM of mIPSC frequency. $n = 17$ neurons for AAV-Vector group and $n = 18$ neurons for AAV-RIIβ-S112A group, *$P < 0.05$, unpaired two-tailed Student's $t$-test. **c** Cumulative probability plots of inter-event interval distributions. Insets: comparison of the representative of an averaged mIPSC. **d** Representative raw recording of miniature excitatory postsynaptic currents (mEPSCs) in DG neurons transfected with AAV-Vector (upper trace, black) and AAV-RIIβ-S112A (lower trace, red). **e** Comparison of the average values ± SEM of mEPSC frequency. $n = 18$ neurons for each group, ns no significance, $P > 0.05$, unpaired two-tailed Student's $t$ test. **f** Cumulative probability plots of inter-event interval distributions. Insets: comparison of the representative of an averaged mEPSC. **g** Representative raw recording of miniature inhibitory postsynaptic currents (mIPSCs) in WT (upper trace, black) and _RIIβ−/−_ (lower trace, blue) DG neurons. **h** Comparison of the average values ± SEM of mIPSC frequency. $n = 15$ neurons for WT group and $n = 16$ neurons for _RIIβ−/−_ group, *$P < 0.05$, unpaired two-tailed Student's $t$ test. **i** Cumulative probability plots of inter-event interval distributions. Insets: comparison of the representative of an averaged mIPSC. **j** Representative raw recording of miniature excitatory postsynaptic currents (mEPSCs) in DG neurons transfected with AAV-Vector (upper trace, black) and AAV-RIIβ-S112A (lower trace, red). **k** Comparison of the average values ± SEM of mEPSC frequency. $n = 16$ neurons for each group, ns no significance, $P > 0.05$, unpaired two-tailed Student's $t$ test. **l** Cumulative probability plots of inter-event interval distributions. Insets: comparison of the representative of an averaged mEPSC. Data were represented as mean ± SEM.

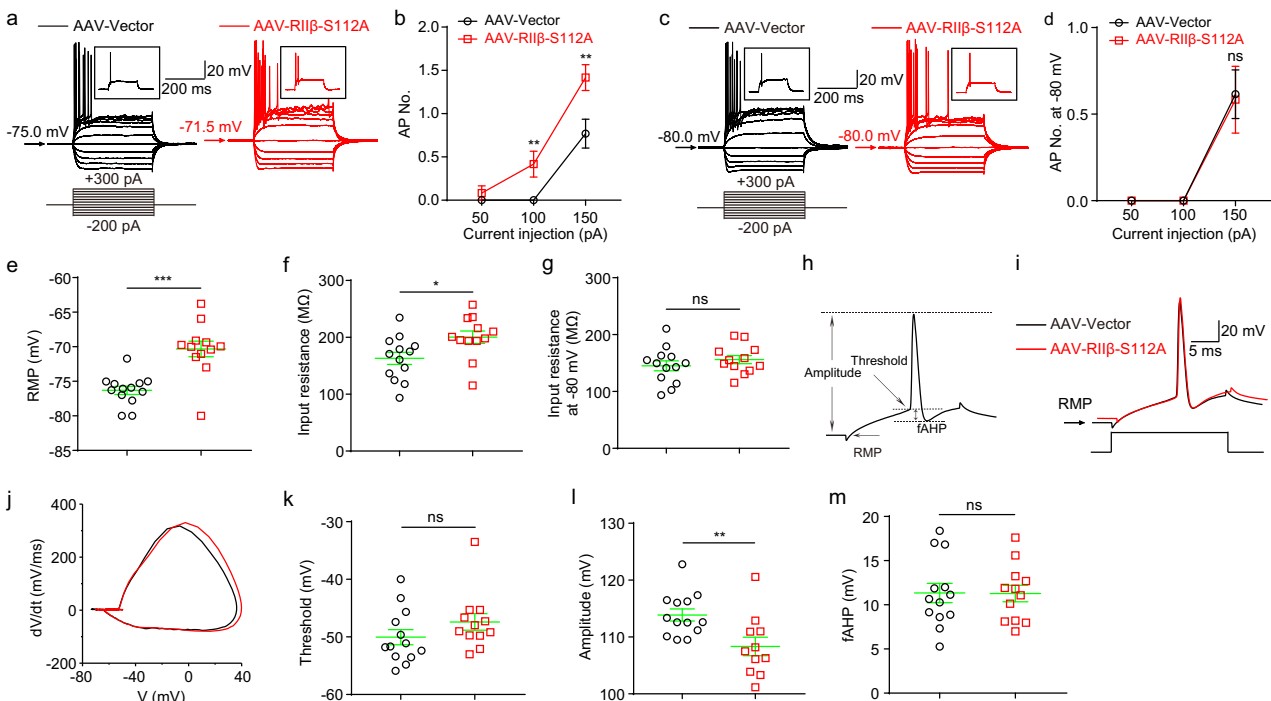

**Fig. 5 Downregulation of autophosphorylation of RIIβ in vivo increased neuronal intrinsic excitabilities of DG granule cells in mice. a** Representative current-clamp recordings of DG neurons held at the normal RMP from AAV-Vector transfected (black) and AAV-RIIβ-S112A transfected (red) mice. A series of 400-ms hyperpolarizing and depolarizing steps in 50-pA increments were applied to produce the traces. Inset: representative trace in response to 150 pA positive current injection. **b** Mean number of action potentials (APs) generated in the response of depolarizing current pulses at RMP. $n = 13$ neurons for AAV-Vector group and $n = 12$ neurons for AAV-RIIβ-S112A group, **$P < 0.01$, unpaired two-tailed non-parametric Mann–Whitney $U$-test for each current pulse. **c** Representative current-clamp recordings of DG neurons held at a fixed potential of −80 mV from AAV-Vector transfected (black) and AAV-RIIβ-S112A transfected (red) mice. A series of 400-ms hyperpolarizing and depolarizing steps in 50-pA increments were applied to produce the traces. Inset: representative trace in response to 150 pA positive current injection. **d** Mean number of APs generated in the response of depolarizing current pulses at −80 mV. $n = 13$ neurons for AAV-Vector group and $n = 12$ neurons for AAV-RIIβ-S112A group, ns no significance, $P > 0.05$, unpaired two-tailed non-parametric Mann–Whitney $U$-test for each current pulse. **e** Individuals and mean spike RMP values. ***$P < 0.001$, unpaired two-tailed Student's $t$ test. **f** Individuals and mean input resistance values at RMP. *$P < 0.05$, unpaired two-tailed Student's $t$ test. **g** Individuals and mean input resistance values at −80 mV. ns no significance, $P > 0.05$, unpaired two-tailed Student's $t$ test. **h** Plot of a typical action potential showed its various phases as the action potential passes a point on a cell membrane. **i** Typical spikes of DG neurons from AAV-Vector transfected (black) and AAV-RIIβ-S112A transfected (red) mice at the normal RMP. **j** Associated phase plane plots. **k–m** Individuals and mean spike threshold, amplitude, and fAHP values. ns no significance, $P > 0.05$; **$P < 0.01$, unpaired two-tailed Student's $t$ test. Data were represented as mean ± SEM.

exists in a phosphorylated state due to the rapid autophosphorylation. When intracellular levels of cAMP rise, the phosphorylated and de-phosphorylated RIIβ homodimer start to associate with cAMP and thus releases the active C subunits and increase the PKA activity. Conversely, when cAMP levels fall, the phosphorylated and de-phosphorylated RIIβ can reassociate with C to reconstitute into an inactive tetrameric holoenzyme[31,33,59,60]. It was reported that dephosphorylated RIIβ can associate with the C subunit at least 5 times more rapidly compared with the association rate of phosphorylated RIIβ with

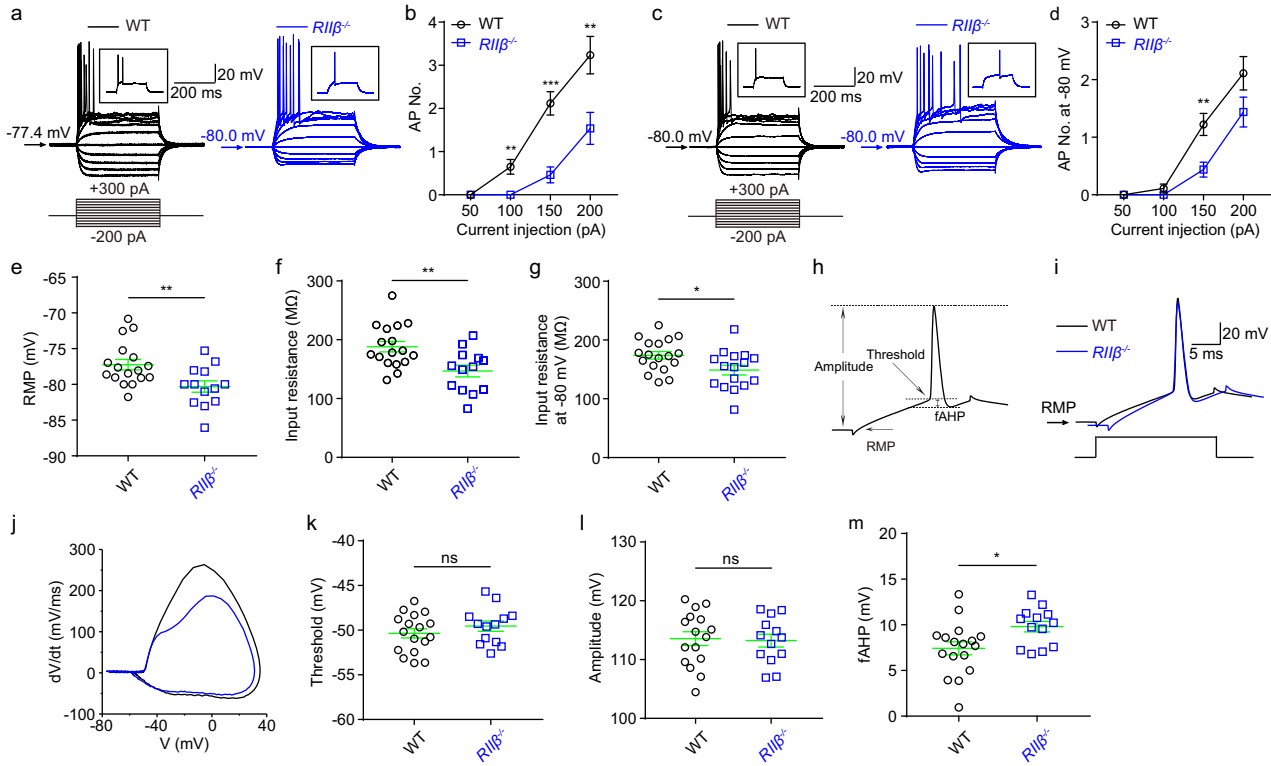

**Fig. 6 Decreased neuronal intrinsic excitabilities of DG granule cells at normal RMP in *RIIβ−/−* mice. a** Representative current-clamp recordings of DG neurons held at the normal RMP from WT (black) and *RIIβ−/−* (blue) mice. A series of 400-ms hyperpolarizing and depolarizing steps in 50-pA increments were applied to produce the traces. Inset: representative trace in response to 150 pA positive current injection. **b** Mean number of action potentials (APs) generated in the response of depolarizing current pulses at RMP. $n = 17$ neurons for WT group and $n = 13$ neurons for *RIIβ−/−* group, **$P < 0.01$; ***$P < 0.001$, unpaired two-tailed non-parametric Mann–Whitney *U*-test for each current pulse. **c** Representative current-clamp recordings of DG neurons held at a fixed potential of −80 mV from WT (black) and *RIIβ−/−* (blue) mice. A series of 400-ms hyperpolarizing and depolarizing steps in 50-pA increments were applied to produce the traces. Inset: representative trace in response to 150 pA positive current injection. **d** Mean number of APs generated in the response of depolarizing current pulses at −80 mV. $n = 18$ neurons for WT group and $n = 16$ neurons for *RIIβ−/−* group, **$P < 0.01$, unpaired two-tailed non-parametric Mann–Whitney *U*-test for each current pulse. **e** Individuals and mean spike RMP values. **$P < 0.01$, unpaired two-tailed Student's *t*-test. **f** Individuals and mean input resistance values at RMP. **$P < 0.01$, unpaired two-tailed Student's *t*-test. **g** Individuals and mean input resistance values at −80 mV. *$P < 0.05$, unpaired two-tailed Student's *t*-test. **h** Plot of a typical action potential showed its various phases as the action potential passes a point on a cell membrane. **i** Typical spikes of DG neurons from WT (black) and *RIIβ−/−* (blue) mice at the normal RMP. **j** Associated phase plane plots. **k–m** Individuals and mean spike threshold, amplitude, and fAHP values. ns no significance, $P > 0.05$; *$P < 0.05$, unpaired two-tailed Student's *t* test. Data were represented as mean ± SEM.

C subunit[59]. This means the dephosphorylated RIIβ subunits can associate with the C subunit more efficiently when compared with phosphorylated RIIβ. Alternatively, the dephosphorylated RIIβ can also be rapidly dephosphorylated by PPs, thus, greatly facilitating its capacity to reassociate with C and generate the inactive holoenzyme. Therefore, the two detectable forms by Western blot analysis are composed of p-RIIβ mainly derived from phospho-holoenzyme, and the disassociated RIIβ (subtracted by p-RIIβ from total RIIβ), which in theory approximately equals to the amount of the disassociated C subunits, in positive proportion to the PKA activity. However, under TLE condition, the abnormal neuronal activities might increase cAMP levels, reduce the RIIβ subunit's autophosphorylation or increase the PPs activity, and hence decrease the proportion of phospho-holoenzyme (detected as p-RIIβ in Western blot analysis), thus to destabilize the holoenzyme, resulting in more liberated C subunits, which is reflected by the disassociated RIIβ subunits. These results suggest that a higher proportion of RIIβ subunits staying in the dephosphorylated states correspond to a higher PKA activity state.

The inhibitory sites of RII subunits are well preserved in both RII isoforms and it is likely impossible to discriminate between p-RIIα and p-RIIβ. The Western blot results we obtained

demonstrated that the RIIβ and p-RIIβ signals were absent from the hippocampal lysates of *RIIβ* knockout mice (Supplementary Fig. 2f), indicating the specificity of the p-RIIβ and RIIβ antibodies in the hippocampal tissues. Two weeks after induced acute seizures, we found that the autophosphorylation level of RIIβ (Ser112) was significantly decreased in both hippocampus (DG, CA1, CA3 regions) and EC, whereas the total RIIβ protein expression was unchanged (Fig. 1h–j), suggesting SE significantly altered the PKA activities without affecting the expression level of RIIβ. Consistent with previous findings, the phosphorylation level at the Ser133 of CREB was increased and the total expression had no significant changes (Fig. 1k, l)[37–40]. Moreover, we confirmed the remarkable enhancement of PKA activities in DG, CA1, CA3, and EC regions in the KA-treated mice model by recognizing the phosphorylated substrates of PKA (Fig. 1m, n). Taken together, our data further confirmed the elevation of PKA activities was linked to the onset of the mouse experimental seizures and RIIβ autophosphorylation might play an important role in the progression of temporal lobe seizures through regulating total PKA activities.

As epileptic seizures are closely related to the neuronal activities[49], our immunostaining results showed that RIIβ subunits are

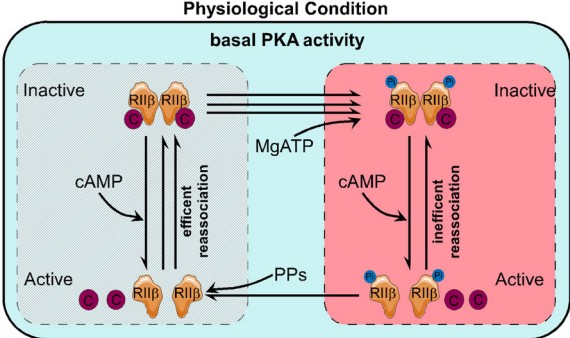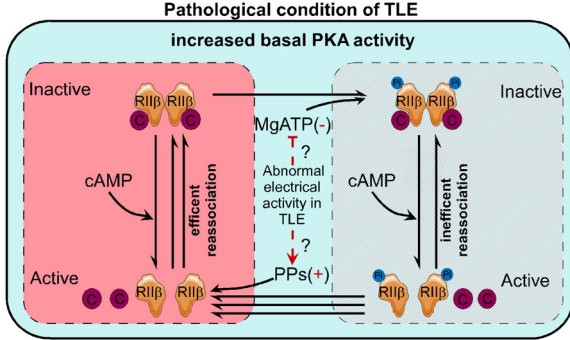

**Fig. 7 The proposed mechanism underlying the increased PKA activities in TLE patients and KA-induced experimental seizure model mice.** In physiological conditions, the PKA-C subunits can autophosphorylate RIIβ subunits using MgATP, and the phosphorylated RIIβ can turn back into unphosphorylated RIIβ using protein phosphatases (PPs). The PKA holoenzyme exists in a phosphorylated state due to the rapid autophosphorylation. When intracellular levels of cAMP rise, the phosphorylated and de-phosphorylated RIIβ homodimer start to associate with cAMP and thus releases the active C subunits and increase the PKA activity. Conversely, when cAMP levels fall, the phosphorylated and de-phosphorylated RIIβ can reassociate with C to reconstitute into an inactive tetrameric holoenzyme[31,33,59,60]. It was reported that dephosphorylated RIIβ can associate with the C subunit at least 5 times more rapidly compared with the association rate of phosphorylated RIIβ with C subunit[59]. This means the dephosphorylated RIIβ subunits can associate with the C subunit more efficiently when compared with phosphorylated RIIβ. Alternatively, the dephosphorylated RIIβ can also be rapidly dephosphorylated by PPs, thus, greatly facilitating its capacity to reassociate with C and generate the inactive holoenzyme. Therefore, the two detectable forms by Western blot analysis are composed of p-RIIβ mainly derived from phospho-holoenzyme, and the disassociated RIIβ (subtracted by p-RIIβ from total RIIβ), which in theory approximately equals to the amount of the disassociated C subunits, in positive proportion to the PKA activity. However, under TLE condition, the abnormal neuronal activities might increase cAMP levels, reduce the RIIβ subunit's autophosphorylation or increase the PPs activity, and hence decrease the proportion of phospho-holoenzyme (detected as p-RIIβ in Western blot analysis), thus to destabilize the holoenzyme, resulting in more liberated C subunits, which is reflected by the disassociated RIIβ subunits. These results suggest that a higher proportion of RIIβ subunits staying in the dephosphorylated states correspond to a higher PKA activity state.

rarely localized to astrocytes (Supplementary Fig. 1b). Our results are consistent with previous findings which showed that RIIβ mRNA levels were abundant in primary neuronal cultures, but were undetectable in primary glial cultures[61,62]. Thus, the neurons seem to be the main workplace for the RIIβ-PKA activities, considering that RIIβ is abundantly expressed in neurons[50]. Therefore, although the lysates contain proteins from non-neuronal cells, the decreased p-RIIβ signal obtained by Western blot was mainly due to a specific downregulation of RIIβ autophosphorylation in neurons.

The reduced CREB phosphorylation in the hippocampus may not be related to a direct function of RIIβ. Because CREB can be phosphorylated by multiple Ca²⁺-activated kinases including PKA, Rsk1, Rsk2, CaMK, MAPKAP, p70 S6 kinase, and PKC[63]. Thus, several signaling cascades are leading to the phosphorylation of CREB. However, it is well established that the phosphorylated PKA substrates (CREB), at least for now, is one of the most powerful tools for investigating the regulation of phosphorylation by PKA; under RIIβ knockout condition, the levels of phosphorylated PKA substrates and CREB indeed decreased drastically, and thus we speculate that RIIβ knockout leads to a dramatic reduction in total PKA activities in the hippocampus.

It was previously reported that an abnormal elevation of PKA activities contributes to an impairment of GABAA receptor function[16]. We studied the effect of autophosphorylation of RIIβ on synaptic transmissions in acute DG slices. We found the frequency of mIPSC recorded from neurons transfected with AAV-RIIβ-S112A was decreased (Fig. 4a–c and Supplementary Table 2), while the mEPSC was not significantly affected (Fig. 4d–f and Supplementary Table 3). Comparable results were also obtained from RIIβ⁻/⁻ slices, the frequency of mIPSC was increased compared with WT slices and mEPSC was unchanged observably (Fig. 4g–l and Supplementary Tables 4 and 5). These results indicated that the downregulation of autophosphorylation of RIIβ decreased inhibitory afferent and RIIβ null increased inhibitory afferent onto DG granule cells. According to the previous findings, enhanced cAMP activities could result in a global

increase in the neural circuit excitability and memory by decreasing inhibitory synaptic transmissions[57]. Our results support the conclusion that PKA suppresses GABAergic synaptic transmissions[64] by regulating postsynaptic GABA receptor sensitivity through phosphorylation[57,65] even in DG granule cells. In sum, the reduced autophosphorylation of RIIβ in hippocampus DG granule neurons in vivo enhanced network excitabilities. We also found decreased network excitabilities when recording CA1 pyramidal neurons, though details shown in mIPSC and mEPSC were different from DG cells (Supplementary Tables 10 and 11). Notably, our mEPSC result obtained from CA1 (Supplementary Table 11) was also in agreement with the previous findings that activating the cAMP/PKA-dependent signaling pathway enhanced the mEPSC frequency but not amplitude in hippocampal CA1 neurons[66]. This differential between mIPSC and mEPSC could be due to the brain region-specific phosphorylation of glutamate receptors and GABAA receptors, and thus produces a region-specific modulation of network activities[67].

The transduction with the AAV-RIIβ-S112A may lead to the expression of RIIβ in astrocytes due to the unspecific CMV promoter. However, the transduced florescence by AAV9-RIIβ-S112A was mainly localized in the cells especially within the dentate gyrus granule cell layer (Supplementary Fig. 2c). Our result is consistent with a previous study[68], which demonstrated a relative specificity for the transduction pattern of the CMV promoter via AAV9 vector by I) AAV9-CMV transduction incorporating cells with neuronal morphology in the hippocampal subregions among different AAV vectors, II) and overall strength of the fluorescence signal on sections matching the Western and biophotonic data. Considering the absence of endogenous RIIβ in astrocytes, our opposite results obtained from RIIβ⁻/⁻ mice were thus mainly from neurons rather than astrocytes. Combining all these data, we did not further apply the neuronal-specific promoter to control the expression of AAV-RIIβ-S112A.

The RIIβ knockout model used in this study is a feasible tool to study decreased PKA activities related mechanisms. We have verified that the global RIIβ knockout mice are fertile and long-

lived, exhibiting no overt abnormal phenotype[69,70]. At present, it is reported that, in the hippocampus, excitatory synaptic transmission (assessed by input/output relations), presynaptic function (assessed by paired-pulse facilitation), and protein synthesis-independent, NMDAR-dependent LTP were normal in the hippocampus of global *RIIβ* knockout mice[71]. Besides, global knockout of *RIIβ* in mice is associated with a compensatory increase of the RIα isoform in various tissues, such as the hypothalamus, brown adipose tissue, white adipose tissue, etc[51,72]. Thus, although the PKA activity was decreased in *RIIβ* KO mice, the hippocampus of global *RIIβ* knockout mice is still intact.

It has revealed significant differences in the ratio of type I (RI-containing) to type II (RII-containing) holoenzyme by analyzing a variety of mammalian tissues. In mice, brain and adipose tissues contain principally the type II holoenzyme[73–75]. In adipocytes, the RIIβ subunits preferentially associate with C, leaving a pool of free RIα that is rapidly degraded. Type I holoenzyme is only formed when the level of C subunits exceeds the level of RII subunits (in this case caused by the loss of RIIβ). Besides, global knockout of *RIIβ* in mice is associated with a compensatory increase of the RI isoform in various tissues, such as the hypothalamus, brown adipose tissue, white adipose tissue, etc[51,72]. In this situation, RIα can successfully compete for binding to the pool of free C subunits and is therefore stabilized in a holoenzyme complex. However, the increases in these R subunits did not compensate fully for the loss of RIIβ, for example, there is a 30% loss of R subunits overall in mutant white adipose tissue, as assessed by total cAMP-binding capacity[76], leaving the unbound C subunits susceptible to proteolysis. Thus, the catalytic subunits of PKA (Cα and Cβ) were reduced dramatically, as predicted by the decreased level of p-CREB and p-substrates of PKA.

The neurons transfected with AAV-RIIβ-S112A have more depolarized RMP values, which is the main factor for increased neuronal excitabilities (Fig. 5), as these neurons fire the same number of APs as control neurons when they were artificially held at a fixed potential of −80 mV (Supplementary Fig. 3). Consistently, *RIIβ*[−/−] neurons have hyperpolarized RMPs and decreased neuronal intrinsic excitabilities (Fig. 6 and Supplementary Fig. 4). It is reported that membrane-residing PKA substrates are preferentially phosphorylated by PKA catalytic subunits compared with cytosolic substrates[77]. And the neuronal membrane-localized ion channels including Nav1.1[12], Kv7[78,79], ROMK1[80,81], L-type Ca$^{2+}$ channels[78,79], AMPA[82,83], NMDA[35], GABA$_A$[14,16] receptors and gap-junctions[11] etc, are regulated by PKA phosphorylation. The possible reason for the changed RMP may be caused by those ion channels regulated by the altered PKA signal activities. However, the differences found in AP firings and input insistence values as well as fAHP at RMP, remained only in *RIIβ*[−/−] rather than AAV-RIIβ-S112A transduced group when cells were artificially held at a fixed potential of −80 mV (Supplementary Fig. 4), thus these changes may be due to the developmental changes for neuronal intrinsic properties[84–86], either by alterations in ion channels caused by PKA subunits compensatory for RIIβ deficiency or non-PKA compensatory mechanism. It seems that the changes in AP amplitude, found either in AAV-RIIβ-S112A transduced neurons only holding at the RMP but not at −80 mV (Fig. 5l and Supplementary Fig. 3e) or in *RIIβ*[−/−] neurons only holding at −80 mV but not at the RMP (Fig. 6l and Supplementary Fig. 4e), is thus unlikely mediated by alterations in ion channels due to direct changes in PKA-RIIβ. Recently, Tiwari et al.[87] found that in hippocampus CA1 neurons from TLE rats, PKA-mediated suppression of the slow afterhyperpolarization (sAHP), which was induced by downregulation of KCa3.1. In our study, we could also find similar results that the enhanced PKA activities by reduced autophosphorylation of RIIβ in vivo could lead to increased neuronal firings with an elevated sAHP (Fig. 5i and Supplementary Fig. 3b, statistical results not shown), while the

decreased PKA activities in *RIIβ* null mice showed reduced neuronal firings and a trend for lower sAHP (Fig. 6i and Supplementary Fig. 4b, statistical results not shown). Thus, the sAHP-KCa3.1 current might also be mediated by the autophosphorylation of RIIβ, which is consistent with the previous study[87].

Overall, in the present study, we identified that the dynamic autophosphorylation level of the PKA-RIIβ subunit was critically involved in both intrinsic and network excitabilities. Given that both intrinsic and network excitabilities play essential roles during normal postnatal development and learning and memory, as well as in various chronic neurological and psychiatric disorders[6,88], our results suggested that modulating PKA activities via the autophosphorylation of RIIβ subunit might play a key role in both physiological and pathological processes in the brain.

## Methods

**Reagents and antibodies**. Drugs used were as follows kainic acid monohydrate (from Milestone PharmTech, 30 mg/kg), N-(2-aminoethyl) biotin amide hydrochloride (NEUROBIOTIN™ Tracer, SP-1120, from Vector Laboratories), Alexa 488-conjugated streptavidin (Molecular Probes, Eugene, OR, USA), tetrodotoxin (form Baomanbio, sodium channel blocker, 500 nM, stock concentration of 500 μM in pH 4.8 citrate buffer), AP5 (from Sigma, NMDA receptor antagonist, 50 μM, stock concentration of 50 mM in pure water), CNQX (from Sigma, AMPA/kainate glutamate receptor antagonist, 10 μM, stock concentration of 10 mM in DMSO), bicuculline (from Abcam, GABA$_A$ receptor antagonist, 10 μM, stock concentration of 10 mM in pure water), muscimol (from Abcam, GABA$_A$ receptor agonist, 1 μM, stock concentration of 1 mM in pure water). Commercial antibodies used were: Anti-PKA RIIβ (pS114) (BD 612550, 1:1000), Anti-PKA RIIβ (BD 610625, 1:1000), Anti-Phospho-CREB (Ser133) (CST #9198, 1:1000), Anti-CREB (CST #9197, 1:1000), Anti-Phospho-PKA Substrate (CST #9624, 1:1000), Anti-GAPDH (Sigma G8795, 1:10000), Anti-β-actin (Sigma A5316, 1:10000).

**Animals**. Male C57BL/6 mice (18–20 g) of SPF grade were obtained from Vital River Laboratory Animal Technology Co, Ltd. (Permit number: SCXK 2012-0001). *RIIβ* knockout mice were provided by Dr. G. Stanley McKnight (University of Washington). Mice carrying the conditional RIIβ allele were generated by inserting a loxP-flanked neomycin resistance gene (neo-STOP) into the RsrII site ∼50 bp upstream of the RIIβ ATG codon. Using standard ES cell procedures, germ-line–transmitting chimeric animals were obtained and then backcrossed with C57BL/6 for at least six generations[50]. All animals were kept on a 12-h light/12-h dark regimen, with free access to food and water. We performed the experimental procedures following the National Institutes of Health Guide for the Care and Use of Laboratory Animals and the procedures were approved by the Biomedical Ethics Committee for animal use and protection of Peking University. Each effort was made to minimize animal suffering and the number of animals used. The experiments were blind to viral treatment or drug treatment conditions during behavioral testing.

**Genotyping**. Offspring's tail genomic DNA was extracted and genotyped using PCR analysis. Briefly, genomic DNA was extracted from 0.2 cm tail snips using the alkali extraction method[89]. A PCR reaction was then performed using Ex Taq polymerase (TaKaRa) and the following primers: (1) RIIβ-Base: 5′-AGGAGCTGG AGATGCTGCCAA-3′; (2) RIIβ-WT: 5′-TCAGCACCTCCACCGTGAA-3′; (3) RIIβ-KO: 5′-GTGGTTTGTCCAAACTCATCAATGT-3′. Primers (1) and (3) were used to detect the *RIIβ* knockout allele (220 bp), while primers (1) and (2) detect the WT RIIβ allele (260 bp). For each sample, a total volume of 20 μL PCR reaction was prepared by mixing the following ingredients: 2 μL of 20 μM primer (1), 2 μL of 20 μM primer (2), 2 μL of 20 μM primer (3), 10 μL 2X Taq PCR MasterMix, 2 μL ddH$_2$O and 2 μL DNA sample. The PCR reaction protocol consisted of an initial 5 min at 95 °C, followed by 37 cycles of 30 s at 95 °C, 20 s at 65 °C, and 30 s at 72 °C. After the last cycle, the reaction is kept at 72 °C for 10 min and then held at 4 °C. A 260 bp band was observed for mice containing the WT allele, and a 220 bp band was seen for mice containing the mutant allele and heterozygotes contained both alleles.

**Control tissues or tissues with TLE**. Patients ($n = 355$) with medically intractable TLE underwent phased presurgical assessment at Shengjing Hospital affiliated to China Medical University. Epilepsy diagnosis (including types and localization) as determined by clinical history, imaging examination (including MRI and/or PET), EEG (including the scalp and/or intracranial EEG), and psychological assessment. Patients with TLE caused by stroke, tumor, injury, and malformations were excluded from this study. In those selected for surgery, the epileptic foci were resected according to standard procedures. Between July 2010 and February 2016, 246 hippocampi were resected. The study using clinical samples, which include 14 paired epileptogenic tissues and matched adjacent normal tissues, was approved by the Ethics Committee of Shengjing Hospital affiliated to China Medical University (Supplementary Table 1). Tissues were frozen in liquid nitrogen immediately after

surgical removal and maintained at −80 °C until protein extraction. Informed consent was obtained from all subjects or their relatives.

**Acute slice preparations**. Adult C57BL/6 mice (3–4 months old) were deeply anesthetized with sodium pentobarbital (60 mg/kg, i.p.) and then transcardially perfused, and decapitated to dissect brains into ice-cold slicing solution containing the following (in mM): 110 Choline chloride, 2.5 KCl, 1.25 $NaH_2PO_4$, 25 $NaHCO_3$, 0.5 $CaCl_2$, 7 $MgCl_2$, 25 glucose, 0.6 Sodium ascorbate, 3.1 Sodium pyruvate (bubbled with 95% $O_2$ and 5% $CO_2$, pH 7.4). Acute horizontal hippocampal slices (350-μm in thickness) were cut by using a vibratome (Leica VT1200S, Germany) and transferred to normal artificial cerebrospinal fluid (aCSF) (in mM): 125 NaCl, 2.5 KCl, 2.0 $CaCl_2$, 2.0 $MgCl_2$, 25 $NaHCO_3$, 1.25 $NaH_2PO_4$, 10 glucose (bubbled with 95% $O_2$ and 5% $CO_2$, pH 7.4). Then, slices were incubated at 37 °C for 20–30 min and stored at room temperature before use.

**In vitro electrophysiological whole-cell patch-clamp recordings**. All somatic whole-cell patch-clamp recordings were performed from identified hippocampal DG granule neurons and CA1 pyramidal cells[89]. For whole-cell current-clamp recordings, the internal solution contained (in mM): 118 $KMeSO_4$, 15 KCl, 2 $MgCl_2$, 0.2 EGTA, 10 HEPES, 4 $Na_2ATP$, 0.3 Tris-GTP, 14 Tris-phosphocreatine, adjusted to pH 7.3 with KOH. Bicuculline, CGP55845, AP5 and CNQX were applied to block inhibitory and excitatory synaptic transmissions. The input resistance was calculated with the equation:

$$\text{Input resistance} = \left( V_{\text{baseline}} - V_{\text{steady-state}} \right) \times 10 \ (\text{M}\Omega) \quad (1)$$

Where $V_{\text{baseline}}$ is the normal resting membrane potential or −80 mV, and $V_{\text{steady-state}}$ is the voltage recorded at 0–10 ms before the end of the −100 pA stimulus.

For whole-cell voltage-clamp recordings of mEPSCs, the internal solution was the same as used in current-clamp recordings. TTX, bicuculline, and CGP55845 were applied to block inhibitory synaptic transmissions.

For whole-cell voltage-clamp recordings of mIPSCs, the internal solution contained (in mM): 118 CsCl, 1 $CaCl_2$, 5 $MgCl_2$, 10 EGTA, 10 HEPES, 4 $Na_2ATP$, 0.3 Tris-GTP, 14 Tris-phosphocreatine, adjusted to pH 7.3 with CsOH. TTX, AP5, and CNQX were applied to block excitatory synaptic transmissions.

We used thick-wall borosilicate pipettes with open-tip resistances of 4–6 MΩ. All recordings were started at least 10 min after breakthin for internal solution exchange equilibrium and performed with a MultiClamp 700B amplifier (Molecular Device) and data were acquired using pClamp 10.6 software either at the normal RMP or at a fixed potential of −80 mV, filtered at 2 kHz and sampling rate at 33 kHz with a Digidata 1440 A digitizer (Molecular Devices). Slices were maintained under continuous perfusion of aCSF at 32–33 °C with a 2–3 mL/min flow. In the whole-cell configuration series resistance (Rs) 15–30 MΩ, and recordings with unstable Rs or a change of Rs >20% were aborted.

For cell labeling, the internal solution contains 0.1–0.2% (w/v) neurobiotin tracer. At the end of the electrophysiological recording of the recorded neurons (about 30 min), slices were treated as previously described[90]. Briefly, sections were fixed in 4% paraformaldehyde in 0.1 M phosphate buffer (pH 7.4) for 20–30 min at room temperature, and subsequently washed 3–4 times for 30 min in 0.1 M phosphate-buffered saline pH 7.4 (PBS) at 4 °C. Sections were then incubated in Alexa 488-conjugated streptavidin (overnight 4 °C, 1: 250 in blocking solution) to visualize neurobiotin.

**Kainic acid-induced temporal lobe seizures**. Seizure induction was performed as previous studies[48,91,92]. Adult mice (3–4 months old) were administered kainic acid (30 mg/kg, i.p.). Animals were observed continuously for 2 h or 2.5 h after kainic acid administration, and seizure behaviors were recorded with the time of onset in minutes from the injection. Kainic acid-induced seizures were rated by a modified scale as described[93]: (1) Immobility; (2) Tremors, not continuous and/or with one forepaw liftoff; (3) Continuous body tremor and/or with body upright standing by hind paws; (4) Rearing, limb clonus with loss of postural control, and/or bouncing seizures; (5) Tonic hindlimb extension or death. Animals were considered fully induced to status epilepticus (SE), which (usually begins with three consecutive Class 4 (generalized) seizures, lasting for at least 30 min[48,94]. SE will be terminated at 1 h after the first Class 4 seizure occurs or at the end of behavior monitoring if the maximum seizure could not reach Class 4 with the use of commercial sodium pentobarbital (30 mg/kg, i.p.). For Western blot analysis, those experimental KA-injected mice, which could not develop into SE at 90 min after injection, were administrated another dose of KA (10 mg/kg, i.p.) to result in SE finally. The control mice were treated with normal saline (i.p.) and pentobarbital sodium (30 mg/kg, i.p.).

**Sterotaxically guided AAV injection**. AAVs used in this study (AAV9-CMV-bGlobin-MCS-3FLAG-2A-EGFP, AAV9-CMV-bGlobin-MCS-3FLAG-2A-RIIβ-S112A) were purchased from Genechem Co. Ltd. For surgeries, animals of 2 months old were deeply anesthetized by intraperitoneal injection of sodium pentobarbital (50 mg/kg of body weight) and secured in the stereotaxic apparatus with ear-bars (RWD Ltd, China). After exposing the skull via a small incision, two small holes were drilled for injection based on coordinates to bregma. A syringe needle with 200-μm diameter was inserted into hippocampal DG regions (medial/lateral: ±1.00 mm; anterior/posterior: −1.50 mm; dorsal/ventral: 2.00 mm below the dura), respectively, and AAV

viruses (500 nL per injection site) were bilaterally injected at a rate of 50 nL/min in an in-house-built air-puff system. The needle was left in position for another 10 min to allow enough absorption and spreading of AAVs before being withdrawn. Animals were allowed to recover from surgery for 1 week and their body weight and health conditions were closely monitored during recovery. Since AAV was strongly expressed after 7 days post-injection (DPI) and reach a steady-state expression level at 2 weeks after injection, the tested mice were used on 21 DPI.

**Total protein extraction and Western blotting analysis**. For quantification of protein expression, mice were deeply anesthetized with sodium pentobarbital sodium (60 mg/kg, i.p.) and decapitated 2 weeks after administration of KA or saline. Acute horizontal hippocampal slices containing dorsal hippocampus in 350-μm thickness were cut and collected by using a vibratome (Leica VT1200S, Germany). Medial EC, CA1, CA3a-b, and DG subfields were rapidly micro-dissected and stored at −80 °C until use based on previous studies[95,96]. Brain tissues from mice and patients were homogenized with a Polytron in ice-cold RIPA buffer [1% Triton X 100; 10 mM $Na_2HPO_4$ (sodium phosphate); 150 mM NaCl (sodium chloride); 1% DOC (Sodium deoxycholate or deoxycholic acid); 5 mM EDTA; 5 mM NaF (sodium fluoride); 0.1 % SDS] supplemented with protease and phosphatase inhibitors (catalog #P8340 and #P2850; Sigma), sonicated and cleared by centrifugation (10,000 × $g$, 10 min, at 4 °C). Protein concentration in the supernatant was determined by BCA assay (Aidlab; PP01). Protein (5 μg for mice, 20 μg for patients) in 1 × sample buffer [62.5 mM Tris-HCl (pH 6.8), 2% (wt/vol) SDS, 5% glycerol, 0.05% (wt/vol) bromophenol blue] was denatured by boiling at 100 °C for 5 min and separated on 8% sodium dodecyl sulfate-polyacrylamide (SDS-PAGE) gels and transferred onto a nitrocellulose membrane (Pall Corporation; T60327) by electrophoresis. Blots were blocked in 5% nonfat milk in Tris-buffered saline and Tween 20 (TBST) for 2 h at room temperature and probed with the primary antibody in 5% BSA-TSBT overnight at 4 °C. After overnight incubation, the blots were washed three times in TBST for 15 min, followed by incubation with HRP-conjugated secondary antibody in TBST with 5% nonfat milk for 2 h at room temperature. Following three cycles of 15 min washes with TBST, the blots were developed using an Enhanced Chemiluminescence assay (BIO-Rad). Densitometry analysis was performed on scanned Western blot images using the ImageJ software (NIH).

**Frozen section**. Adult mice (3–4 months old) were anesthetized with pentobarbital and brain were immediately dissected and fixed in 4% paraformaldehyde solution for 48 h at 4 °C, then cryoprotected by immersion for 24–48 h in gradient sucrose (10%, 20%, and 30%) with 0.01 mol/L PBS at 4 °C and subsequently frozen in OCT compound (Sakura FineTech, Tokyo). Tissue sections of 20 μm in thickness were taken on a cryostat and allowed to air dry on slides, followed by analysis on a microscope (Histology Facility of Department of Anatomy, Histology and Embryology, Peking University).

**Statistics and reproducibility**. For in vivo experiments, the animals were distributed into various treatment groups randomly. Group data are represented as mean ± SEM. For Western analysis, $n ≥ 5$ mice for each group; for patch-clamp recordings, $n = 3–5$ neurons per mouse with at least 3 mice for each group. GraphPad Prism was used for the analysis of variance. Comparisons between two groups were made using Student's paired or unpaired two-tailed $t$-test, or unpaired two-tailed non-parametric Mann–Whitney $U$-test. Comparisons among three or more groups were made using one-way ANOVA analyses followed by Bonferroni's multiple-comparisons test. Data collection and analysis were not randomized or performed blind to the conditions of the experiments. All data were expressed as the means ± SEM. Statistical significance of differences at $P < 0.05$ is indicated as one asterisk (*); $P < 0.01$ is indicated as two asterisks (**); and $P < 0.001$ is indicated as three asterisks (***) in all figures.

**Reporting summary**. Further information on experimental design is available in the Nature Research Reporting Summary linked to this paper.

## Data availability
All data underlying the main and Supplementary figures are either available online in Supplementary Data 1, or available from the corresponding authors [ZH and RZ], upon reasonable request.

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

## Acknowledgements

We thank Dr. G. Stanley McKnight (University of Washington) for providing us the RIIβ knockout mice. This work was supported by the National Natural Science Foundation of China, Grant Nos. 81371432 (to ZH), 81471064 and 81670779 (to R.Z.); Beijing Natural Science Foundation (Grant No. 7182087 to ZH and 7162097 to R.Z.); Beijing Municipal Science & Technology Commission (Grant No. Z181100001518001 to Z.H.); The Peking University Research Foundation (No. BMU20140366 to R.Z.), National Key Research and Development Program of China (2017YFC1700402 to R.Z.).

## Author contributions

J.Z., C.Z., R.Z., and Z.H. designed the experiments. J.Z., C.Z., X.C., and B.W. performed the experiment and analyzed the data. W.M. provided the patients' tissues and the information. W.M. and Y.Y. participated in data analysis, experimental design and manuscript writing. R.Z. and Z.H. supervised the project. J.Z., C.Z., R.Z., and Z.H. wrote the paper with inputs from all authors.

## Competing interests

The authors declare no competing interests.
