## [Peer Review File · Communications Biology]

Reviewers' comments:

Reviewer #1 (Remarks to the Author):

Brief summary of the manuscript

Zhang et al. investigated the role of PKA-RII β signaling during temporal lobe epilepsy (TLE) in the hippocampus. They found that the level of PKA-RII autophosphorylation within the inhibitory domain was robustly decreased in hippocampal tissue of epileptic foci obtained from TLE patients and in a TLE mouse model. The phosphorylation level of RII subunits was negatively correlated with the overall activities of PKA isoforms determined by analyzing CREB and PKA-substrate phosphorylation. To support this observation, the authors transduced hippocampal cells with an adeno-associated virus expressing a phosphorylation-resistant mutant of RII β (S112A) and analyzed RII β knockout mice. Transduction with AAV-RII β -S112A led to an increase in hippocampal PKA activity. In the contrary, phosphorylation of PKA substrates and CREB was reduced in RII β knockout mice. By electrophysiological recordings, the authors then demonstrate that transduction of hippocampal cells with AAV-RII β -S112A significantly decreased miniature IPSCs (mIPSCs) frequency but not mEPSCs onto hippocampal DG granule cells and enhanced the neuronal intrinsic excitability and seizure susceptibility. In RII β knockout mice, the authors observed an increase in mIPSCs frequency onto DG neurons and reduced neuronal excitability and epileptogenesis.

Overall impression of the work

Their study supports previous findings indicating that PKA activity is essential for the pathogenesis of TLE and provides interesting new insights concerning the relevant PKA isoform and mechanism. Overall, the manuscript is clearly written and most of the relevant literature is mentioned. The experiments are well conducted, presented, and the interpretation of most findings is appropriate. The statistical analysis of data is sound. Nevertheless, I have some major concerns that I believe should be addressed prior to publication. Unfortunately, the authors did not address how PKA-RII β is activated during TLE and do not reveal how this leads to reduced RII phosphorylation. In addition, the cellular basis of some of the findings remains unclear.

Specific comments

Major points

Line 114-125 and 149-152: The authors analyzed the phosphorylation of RII β using a phospho-specific antibody by western blotting. The data suggest a robust reduction of RII β phosphorylation in human TLE patients and mice (kainite model). The used antibody is indeed specific for RII inhibitory sites, but cannot discriminate between RII α and RII β (see Isensee et al., J Cell Sci, 2014). The inhibitory sites of RII subunits are well preserved in both RII isoforms and it is likely not possible to discriminate between pRII α and pRII β . The decrease in the pRII signal obtained by western blotting may be due to downregulation of RII α expressed in various neuronal and non-neuronal cell types (see e.g. mousebrain.org). The authors should therefore not only quantify the abundance of RII β , but also RII α in the very same lysates. Moreover, the authors had access to RII β knockout mice. The absence of the pRII signal in hippocampal lysates of RII β knockout mice should prove the specificity of the pRII antibody for RII β in the investigated tissue. I think these experiments are needed to support the claims made in the paper.

Line 114-125 and 149-152: The authors state that the activity of PKA-II is dynamically controlled by a

process of autophosphorylation. Recent studies clearly indicate that RII inhibitory sites are fully phosphorylated in the inactive kinase (Zhang P et al., 2012 and 2015) and acute stimulation with cAMP does not result in changes of the RII phosphorylation level, which would be detectable by western blotting. In fact, pRII antibodies were successfully applied to measure the dissociation and thereby activation of PKA-II in intact neurons by immunocytochemistry (Isensee J et al., 2017, 2018). In cells, increased pRII immunoreactivity reflects increased accessibility of the already phosphorylated RII epitope during cAMP-induced opening of PKA-II. It is interesting to see that activity of neurons during TLE leads to a reduction of RII phosphorylation, but the authors do not explain how this results in increased activity of PKA-II. It could be that long-lasting or repetitive activation of PKA-II leads to accumulation of dephosphorylated RII, or that TLE activates phosphatases that dephosphorylate RII. The authors, however, do not mention or discuss how they think this mechanism might work. In addition, the findings would have been more solid, if they would have quantified the level of RII phosphorylation in hippocampal sections by immunohistochemistry. In the current manuscript, it remains unclear in which cells the regulation of RII phosphorylation actually happens.

Line 170: The authors claim that the level of RII β autophosphorylation was significantly reduced in AAV-RIIb-S112A positive neurons. This is not supported by the data presented in Fig. 2B-C, since hippocampal protein lysates were analyzed by western blotting. These lysates contain protein from non-neuronal cells that were also transduced by the AAV. To support this claim, the authors need to label hippocampal DC sections and quantify fluorescence signals in individual neurons.

Line 177: I am not sure why the authors did not detect an increase in the abundance of RII β after AAV-mediated expression of the S112A mutant of RII β . The used antibody should detect the RII β mutant as well leading to an increase in overall RII β abundance. There are some additional bands in some of the samples shown the blot (Fig. 2B), but they do not correlate with the AAV transduction. Please show the uncropped blot in the revision, just to make sure that nothing was mixed up here.

Line 196: The authors used global RII β knockout mice to demonstrate that lack of RII β reduces the level of phosphorylated CREB and PKA substrates as well as seizure susceptibility. It is known, however, that lack of RII β results in developmental alterations in the brain (PKA function is required for proper dendritogenesis and the organization of cortical layer IV neurons into barrels, Inan et al. 2006). The authors show that lack of RII β results in reduced CREB phosphorylation in various hippocampal regions, which likely translates to CREB dependent gene expression. The interpretation of the findings is difficult and may not be related to a direct function of RII β . Did the authors investigate if the hippocampus of RII β mice is still intact, at least at the level of cellular composition? It might be enough if others have studied that in detail and the authors reference this work.

Minor Points

Line 87: The authors state that mainly RI α , RI β , and RII β are highly expressed in the brain. This might apply to some neurons in the brain, but RI α is certainly also expressed in many neuronal and non-neuronal cells in the brain.

Line 169: Please state if the used AAV drives expression from a neuronal promoter or globally in all cell types of the hippocampal DG area. The methods section states CMV bGlobin-MCS-3FLAG-2A-....., which appears not neuron-specific.

Line 179-182: The sentence structure should be revised.

Line 210: change “a significant less seizure progression” to “significantly less seizure progression”

Line 250: et al., reference is missing.

Line 260: Change “AAV-vector-tranducted” to “AAV-vector-transduced”

Line 346-350: The sentence structure should be revised.

Line 355: “dynamics” to “dynamic”

Figure 1, line 798: Please indicate which tissue has been analyzed and explain abbreviations NC and EP.

Figure 2, line 813: A bit difficult to follow, maybe “Downregulation of RII β autophosphorylation in vivo leads to increased phosphorylation of CREB and PKA substrates as well as increased epileptic seizure susceptibility in mice.”

Figure 2H: Indicate the applied statistic test in the figure legend.

Reviewer #2 (Remarks to the Author):

In this study, the authors addressed the mechanistic role of PKA-RII β subunit (one of the PKA regulatory subunits) in the genesis of epileptic seizures in both human patients and mice. PKA-RII β subunit autophosphorylation was decreased in both human and mouse samples from epileptic hippocampi, and RII β phosphorylation levels negatively correlated with the PKA catalytic activity. KA seizures were increased in mice infused with AAV-RII β -S112A (a persistent phosphorylation form of RII β), while RII β ko mice showed a lower susceptibility to KA seizures. AAV-RII β -S112A transduction in hippocampal neurons decreased miniature IPSCs (mIPSCs) and enhanced the neuronal intrinsic excitability, while increased mIPSCs and reduced neuronal intrinsic excitability was observed in hippocampal cells from RII β ko mice. Taken together, this data indicates that autophosphorylation of RII β subunit is a key player controlling hippocampal excitability.

This is an interesting paper, that likely deserves publication in Communications Biology. However, several parts of the manuscript need clarification and amendments.

Major issues

1) page 6, lines 144-148. The following statement should be modified according to the available literature: "After a delay of a few weeks, known as the latent period (during which animals appear to be normal), spontaneous overt behavioral seizures occur (defined as the onset of chronic TLE). This model is widely used because many of the clinical and pathological features of the human disorder (including the latent period) can be reproduced".

The systemic (intraperitoneal) KA model is known to induce a latent phase followed by chronic epilepsy (recurrent seizures) in the rat (to which the quoted studies refer) but not in the mouse. In addition, the short and long term effects of systemic KA markedly vary across mouse strains (see the

classical studies by P-E. Schauwecker). Specifically, C57BL6 mice are quite resistant to KA and do not develop chronic seizures when given 30mg/kg KA. So, as to my current knowledge, systemic KA cannot be considered a valid model of TLE.

2) In line with the above mentioned issue, the genetic background of RII β ko mice (C57BL6? other?) should be clearly reported in the Methods section.

3) page 8 lines 215-216. Again, the following statement should be amended: "These results suggested that PKA activity in epileptic foci may be causally linked to epileptogenesis". This is not correct; the current study, as designed, is not addressing epileptogenesis, but only the occurrence of acute seizures after KA administration in mice.

4) The Discussion is a mere summary of the results, with minimal interpretation/speculation. Please modify and expand.

Minor issues

1) page 7, line 171. Please better define "in most region of DG": where different cell types transduced by AAV-RII β -S112A ?

2) Methods: please describe how the different hippocampal subfields (DG, CA1, CA3) were dissected for western blot.

3) page 8 lines 223-224. "Hippocampal DG granule cells were identified by its typical..." should read "Hippocampal DG granule cells were identified by their typical..."

Reviewer #3 (Remarks to the Author):

PKA-RII β autophosphorylation modulates the PKA activities and regulates the generation of mouse seizures

In this manuscript Zhang et al., analyze the involvement of the regulatory subunit RII-beta of the protein kinase A, in modulating the generation of seizures. PKA is an important kinase that it is extensively expressed across multiple brain regions and its function is very important to modulate critical physiological functions. PKA is formed by two regulatory subunits and two catalytic. Its activation is mediated for binding of cAMP to the regulatory subunit. The authors pay attention to the role of PKA-RII β subunit as a possible key factor in the regulation of neuronal excitability. They have found a reduction in the autophosphorylation of PKA-RII β , both in human and mouse model of temporal lobe epilepsy. This reduction correlates with an increase in PKA activity and CREB phosphorylation. Complementary results are obtained using different strategies, including the use of KO-PKA-RII β mice or in contrast, via the suppression of the subunit autophosphorylation. Finally, they observe how an increase in PKA activities enhance the neuronal intrinsic excitability in hippocampal DG granule cells and seizure susceptibility. In contrast, a reduction of PKA activities observed in a KO-PKA-RII β can perform a reduction in neuronal intrinsic excitability and less seizure susceptibility.

Overall this is a strong manuscript, but a few areas discussed below need to clarify or improve.

Major comments,

1) In the abstract authors mention that they induce an increase in PKA activities generating a

persistent phosphorylation form of RII β using a transduction with adenovirus. In a deep reading of the manuscript is clear what the authors have made but it is confusing in a first view. A reader could think that it is permanent phosphorylation in the Serine 112 site that mention just above. It would necessary to clarify it in order to understand that it is an inactivation strategy via suppression of autophosphorylation.

2) RII- β is anchored to adenosine kinase anchoring proteins in the cell membrane. This way, RII β subunit regulates the phosphorylation level of neurotransmitter receptors and ion channels. There is a recent article (Tiwari et al, 2019. J.Neurosci) in which describes that the KCa-sAHP reduction in hippocampal neurons in TLE is due to the downregulation of KCa31 channels, mediated by PKA. Could you discard that some of the effects that you find in the excitability are not mediated by alterations in ion channels?

3) The data obtained in the patient samples is really clear, but I am struck that the controls used are adjacent normal tissues. Have you had the opportunity to analyze the status of RII β in the hippocampus of control patients? Could there be differences between areas? In mice model comparison is made with wild types animals in the same areas, have you test the effect in adjacent areas?

4) In Fig.2, authors show as inactivation of autophosphorylation in serine 112 increases the activity of PKA, the rate of phosphorylated CREB protein, the p-PKA substrates, and the seizures. However, there is missing of the manuscript a discussion or mention of the results found using the kainate mice model, concretely the great reduction in p-RII levels and in consequence in the PKA activity (comparison between control and epi of S112A). An explanation of these results would be appropriate to include. Do you think that there are other mechanisms independent of RII-beta? Are these results a consequence of the method used?

5) KO-RII β mice have a decreased level of p-CREB and p-substrates of PKA. The R subunits act as an intrinsic inhibitor of the catalytic subunits but also protect it to degradation. In previous reports, it has described as RII beta deficiency leads to decreased catalytic subunits and PKA activity. I would recommend you include this data in the discussion in view of your results.

Minor comments,

1) Figure legends. I have noted that some information is missing, for example, the explanation of the acronyms used. For example, fig 1A and C, is not described as the meaning of NC or EP. Please review the legend to complete the information.

Point-to-point response to reviewers' comments

We would like to thank the reviewers for taking the time to consider our manuscript carefully and thoughtfully. We appreciate the comments and have done necessary experiments to address these concerns, which we feel have substantially improved and strengthened our manuscript. The specific reviewer's comments are addressed below:

Reviewer #1:

Brief summary of the manuscript:

Zhang et al. investigated the role of PKA-RII β signaling during temporal lobe epilepsy (TLE) in the hippocampus. They found that the level of PKA-RII autophosphorylation within the inhibitory domain was robustly decreased in hippocampal tissue of epileptic foci obtained from TLE patients and in a TLE mouse model. The phosphorylation level of RII subunits was negatively correlated with the overall activities of PKA isoforms determined by analyzing CREB and PKA-substrate phosphorylation. To support this observation, the authors transduced hippocampal cells with an adeno-associated virus expressing a phosphorylation-resistant mutant of RII β (S112A) and analyzed RII β knockout mice. Transduction with AAV-RII β -S112A led to an increase in hippocampal PKA activity. In the contrary, phosphorylation of PKA substrates and CREB was reduced in RII β knockout mice. By electrophysiological recordings, the authors then demonstrate that transduction of hippocampal cells with AAV-RII β -S112A significantly decreased miniature IPSCs (mIPSCs) frequency but not mEPSCs onto hippocampal DG granule cells and enhanced the neuronal intrinsic excitability and seizure susceptibility. In RII β knockout mice, the authors observed an increase in mIPSCs frequency onto DG neurons and reduced neuronal excitability and epileptogenesis.

Overall impression of the work:

Their study supports previous findings indicating that PKA activity is essential for the pathogenesis of TLE and provides interesting new insights concerning the relevant PKA isoform and mechanism. Overall, the manuscript is clearly written and most of the relevant literature is mentioned. The experiments are well conducted, presented, and the interpretation of most findings is appropriate. The statistical analysis of data is sound. Nevertheless, I have some major concerns that I believe should be addressed prior to publication. Unfortunately, the authors did not address how PKA-RII β is activated during TLE and do not reveal how this leads to reduced RII phosphorylation. In addition, the cellular basis of some of the findings remains unclear.

Specific Comments:

Major points:

[Comment 1] *Line 114-125 and 149-152: The authors analyzed the phosphorylation of RII β using a phospho-specific antibody by western blotting. The data suggest a robust reduction of RII β phosphorylation in human TLE patients and mice (kainite model). The used antibody is indeed specific for RII inhibitory sites, but cannot discriminate between RII α and RII β (see Isensee et al., J Cell Sci, 2014). The inhibitory sites of RII subunits are well preserved in both RII isoforms and it is likely not possible to discriminate between pRII α and pRII β . The decrease in the pRII signal*

obtained by western blotting may be due to downregulation of *RII α* expressed in various neuronal and non-neuronal cell types (see e.g. mousebrain.org). The authors should therefore not only quantify the abundance of *RII β* , but also *RII α* in the very same lysates. Moreover, the authors had access to *RII β* knockout mice. The absence of the pRII signal in hippocampal lysates of *RII β* knockout mice should prove the specificity of the pRII antibody for *RII β* in the investigated tissue. I think these experiments are needed to support the claims made in the paper.

[Answer 1] Thanks for the reviewers for these comments and suggestions. By applying *in situ* hybridization technique, Cadd, *et al.* reported that the type RII subunits were expressed in very discrete areas in mouse brain (Cadd & McKnight, 1989). They found that transcripts for *RII α* were found almost exclusively in the media habenula nuclei. Only after long exposures, much lower levels of *RII α* mRNA were detected in the hippocampus, neocortex, piriform cortex, reticular thalamic nuclei and some hypothalamic areas. On the contrary, high mRNA level for *RII β* was present in the hippocampus (Cadd & McKnight, 1989), thus, the pRII signal obtained by western blot was mainly due to a specific downregulation of *RII β* . In addition, we carefully studied the literatures mentioned by the reviewer. Isensee *et al.* used two antibodies, which were anti-phospho-*RII β* , BD, No. 612550 and anti-phospho-RII (S96), Abcam, No. ab32390 (seems to be ab226754 alternatively, based on 'S96' on the website: www.abcam.com). The antibody of anti-phospho-*RII β* , BD, no. 612550 can specifically recognize the human phosphorylated S114 in the *RII β* subunit of PKA and the orthologous phosphorylation site in mouse S112. However, the anti-PKA RII antibody of anti-phospho-RII (S96), Abcam, No. ab32390 (seems to be ab226754 alternatively, based on 'S96' on the website: www.abcam.com) they used, detected both pRII α and pRII β and could not distinguish pRII α from pRII. In our study, we applied anti-phospho-*RII β* (BD 612550) which can specifically distinguish pRII β from pRII. In this way, our results were sufficient to support the conclusion that the autophosphorylation of PKA-*RII β* subunit plays a critical role in controlling neuronal and network excitability in hippocampus by regulating the activity of PKA enzyme.

In fact, in the past twenty years, our lab focused on the function of cAMP/PKA system and was the first successfully isolated and cloned many of the cDNAs for the regulatory and catalytic subunits of the cAMP-PKA. My lab leader, Dr. Stan McKnight, has helped Becton, Dickinson and Company to develop antibody of *RII β* (anti-*RII β* , BD, No. 610625; anti-phospho-*RII β* , BD, No. 612550). By using these antibodies, we investigated the major physiological functions of this signal transduction cascade (Cummings *et al.*, 1996; Zheng *et al.*, 2013).

Cadd G, McKnight GS. Distinct patterns of cAMP-dependent protein kinase gene expression in mouse brain. *Neuron* 3, 71-79 (1989).

Isensee J, *et al.* Pain modulators regulate the dynamics of PKA-RII phosphorylation in subgroups of sensory neurons. *Journal of Cell Science* 127, 216-229 (2014).

Cummings, D. E., Brandon, E. P., Planas, J. V., Motamed, K., Idzerda, R. L., & McKnight, G. S. (1996). Genetically lean mice result from targeted disruption of the *RII β* subunit of protein kinase A. *Nature*, 382(6592), 622-626.

Zheng, R., Yang, L., Sikorski, M. A., Enns, L. C., Czyzyk, T. A., Ladiges, W. C., & McKnight, G. S. (2013). Deficiency of the *RII β* subunit of PKA affects locomotor activity and energy homeostasis in distinct neuronal populations. *Proc Natl Acad Sci U S A*, 110(17), E1631-1640. doi:10.1073/pnas.1219542110.

[Comment 2] *Line 114-125 and 149-152: The authors state that the activity of PKA-II is dynamically controlled by a process of autophosphorylation. Recent studies clearly indicate that RII inhibitory sites are fully phosphorylated in the inactive kinase (Zhang P et al., 2012 and 2015) and acute stimulation with cAMP does not result in changes of the RII phosphorylation level, which would be detectable by western blotting. In fact, pRII antibodies were successfully applied to measure the dissociation and thereby activation of PKA-II in intact neurons by immunocytochemistry (Isensee J et al., 2017, 2018). In cells, increased pRII immunoreactivity reflects increased accessibility of the already phosphorylated RII epitope during cAMP-induced opening of PKA-II. It is interesting to see that activity of neurons during TLE leads to a reduction of RII phosphorylation, but the authors do not explain how this results in increased activity of PKA-II. It could be that long-lasting or repetitive activation of PKA-II leads to accumulation of dephosphorylated RII, or that TLE activates phosphatases that dephosphorylate RII. The authors, however, do not mention or discuss how they think this mechanism might work. In addition, the findings would have been more solid, if they would have quantified the level of RII phosphorylation in hippocampal sections by immunohistochemistry. In the current manuscript, it remains unclear in which cells the regulation of RII phosphorylation actually happens.*

[Answer 2] 1) *“The authors, however, do not mention or discuss how they think this mechanism might work.”* Thanks for your comments. To explain the mechanism underlying the increased PKA activity in TLE patients and in KA-induced experimental seizure model mice, we drew a cartoon as follow to illustrate the possible pathological processes under TLE. In details, the cartoon shows that in physiological condition the PKA-C subunits can autophosphorylate RII β subunits using MgATP, and the phosphorylated RII β can turn back into unphosphorylated RII β using protein phosphatases (PPs). The PKA holoenzyme exists in a phosphorylated state due to the rapidly autophosphorylation. When intracellular levels of cAMP rise, the phosphorylated and de-phosphorylated RII β homodimer start to associate with cAMP and thus releases the active C subunits and increase the PKA activity. Conversely, when cAMP levels fall, the phosphorylated and de-phosphorylated RII β can reassociate with C to reconstitute into an inactive tetrameric holoenzyme (Rangel-Aldao & Rosen, 1976; Ping Zhang et al., 2015; P. Zhang, Kornev, Wu, & Taylor, 2015; P. Zhang et al., 2012). It was reported that dephosphorylated RII β can associate with C subunit at least 5 times more rapidly compared with the association rate of phosphorylated RII β with C subunit (Rangel-Aldao & Rosen, 1976). This means the dephosphorylated RII β subunits can associate with C subunit more efficiently, when compared with phosphorylated RII β . Alternatively, the dephosphorylated RII β can also be rapidly dephosphorylated by PPs, thus, greatly facilitating its capacity to reassociate with C and generate the inactive holoenzyme. Therefore, the two detectable forms by western blot analysis are composed of “p-RII β ” mainly derived from phosphoholoenzyme, and the disassociated “RII β ” (subtracted by p-RII β from total RII β), which in theory approximately equals to the amount of the disassociated C subunits, in positive proportion to the PKA activity. However, under TLE condition, the abnormal neuronal activities might increase cAMP levels, reduce the RII β subunit’s autophosphorylation or increase the PPs activity, and hence decrease the proportion of phosphoholoenzyme (detected as “p-RII β ” in western blot analysis), thus to destabilize the holoenzyme, resulting in more liberated C subunits, which is reflected by the disassociated RII β

subunits. These results suggest that higher proportion of RII β subunits staying in the dephosphorylated states, correspond to higher PKA activity state. Here, we also discussed the possible mechanism in the revised manuscript (see lines 326-352).

2) “In addition, the findings would have been more solid, if they would have quantified the level of RII phosphorylation in hippocampal sections by immunohistochemistry. In the current manuscript, it remains unclear in which cells the regulation of RII phosphorylation actually happens.” Thanks for your suggestion. We have tried many times to quantify the expression level of RII β using immunostaining (as shown in figures below). We found that the RII β subunits distributed uniformly across the brain sections; and RII β subunits are mainly localized to the dendrites and axons of neurons rather than soma. Moreover, we observed that RII β was rarely expressed in astrocytes (see *Comment 3 from reviewer No 1*). Therefore, we applied western blotting to quantify the level of RII β in neurons.

Rangel-Aldao, R. and O.M. Rosen, Dissociation and reassociation of the phosphorylated and nonphosphorylated forms of adenosine 3':5' -monophosphate-dependent protein kinase from bovine cardiac muscle. *Journal of Biological Chemistry*, 1976. 251(11): p. 3375-80.

Zhang, P., et al., Structure and allostery of the PKA RIIbeta tetrameric holoenzyme. *Science*, 2012. 335(6069): p. 712-6.

Zhang, P., et al., Discovery of Allostery in PKA Signaling. *Biophys Rev*, 2015. 7(2): p. 227-238.

Zhang, P., et al., Single Turnover Autophosphorylation Cycle of the PKA RII β Holoenzyme. *PLOS Biology*, 2015. 13(7): p. e1002192.

[**Comment 3**] Line 170: The authors claim that the level of RII β autophosphorylation was significantly reduced in AAV-RII β -S112A positive neurons. This is not supported by the data presented in Fig. 2B-C, since hippocampal protein lysates were analyzed by western blotting. These lysates contain protein from non-neuronal cells that were also transduced by the AAV. To

support this claim, the authors need to label hippocampal DC sections and quantify fluorescence signals in individual neurons.

[Answer 3] To exclude the effect of RII β expression in non-neuronal cells, we have investigated the expression level of RII β in GFAP positive cells using immunostaining. The results show that the expression levels of RII β in GFAP-positive cells are very low, suggesting that RII β subunits are rarely localized to astrocytes (data are shown below). However, RII β is abundantly expressed in neurons (Zheng et al., 2013). Our results are consistently with previous findings which showed that RII β mRNA levels were abundant in primary neuronal cultures, but were undetectable in primary glial cultures (Massa, Walker, Moser, Fellows, & Maurer, 1991; Mucignat-Caretta & Caretta, 2004). Therefore, although the lysates contain protein from non-neuronal cell, the decreased pRII β signal obtained by western blotting was mainly due to a specific downregulation of RII β autophosphorylation in AAV- RII β -S112A positive neurons. We have added these data into

revised manuscript (see **lines 205-211 & 366-374**).

[Comment 4] *Line 177: I am not sure why the authors did not detect an increase in the abundance of RII β after AAV-mediated expression of the S112A mutant of RII β . The used antibody should detect the RII β mutant as well leading to an increase in overall RII β abundance. There are some additional bands in some of the samples shown the blot (Fig. 2B), but they do not correlate with the AAV transduction. Please show the uncropped blot in the revision, just to make sure that nothing was mixed up here.*

[Answer 4] We provided the information below as reviewer's requirement (also see Supplementary Fig. 2D-E).

[Comment 5] Line 196: The authors used global *RIIβ* knockout mice to demonstrate that lack of *RIIβ* reduces the level of phosphorylated CREB and PKA substrates as well as seizure susceptibility. It is known, however, that lack of *RIIβ* results in developmental alterations in the brain (PKA function is required for proper dendritogenesis and the organization of cortical layer IV neurons into barrels, Inan et al. 2006). The authors show that lack of *RIIβ* results in reduced CREB phosphorylation in various hippocampal regions, which likely translates to CREB dependent gene expression. The interpretation of the findings is difficult and may not be related to a direct function of *RIIβ*. Did the authors investigate if the hippocampus of *RIIβ* mice is still intact, at least at the level of cellular composition? It might be enough if others have studied that in detail and the authors reference this work.

[Answer 5] We agree with the reviewer's view that the reduced CREB phosphorylation in the hippocampus may not be related to a direct function of *RIIβ*. Because CREB can be phosphorylated by multiple Ca^{2+} -activated kinases including PKA, Rsk1, Rsk2, CaMK, MAPKAP, p70 S6 kinase, and PKC (Impey et al., 1998). Thus, there are several signaling cascades leading to the phosphorylation of CREB. However, it is well established that the phosphorylated PKA substrates (CREB), at least for now, is one of the most powerful tools for investigating the regulation of phosphorylation by PKA; under *RIIβ* knockout condition, the levels of phosphorylated PKA substrates and CREB indeed decreased drastically, and thus we speculate that *RIIβ* knockout leads to a dramatic reduction in total PKA activity in the hippocampus.

As the reviewer's requirement, we investigated if the hippocampus of *RIIβ*-KO mice was still intact. We have verified that the global *RIIβ* knockout mice are fertile and long-lived, exhibiting

no overt abnormal phenotype (Cummings et al., 1996) (Brandon et al., 1998). At present, it is reported that excitatory synaptic transmission (assessed by input/output relations), presynaptic function (assessed by paired-pulse facilitation), and protein synthesis-independent, NMDAR-dependent LTP were normal in the hippocampus of global *RIIβ* knockout mice (Y. Yang et al., 2009). In addition, global knockout of *RIIβ* in mice is associated with a compensatory increase of the *RIα* isoform in various tissues, such as the hypothalamus, brown adipose tissue, white adipose tissue, etc. (Amieux et al., 1997) (L. Yang & McKnight, 2015). Thus, although the PKA activity was decreased in *RIIβ* KO mice, the hippocampus of global *RIIβ* knockout mice is still intact. We also added this discussion in revised manuscript (see lines 375-384 & 410-420).

Impey S, et al. Cross Talk between ERK and PKA Is Required for Ca²⁺ Stimulation of CREB-Dependent Transcription and ERK Nuclear Translocation. Neuron 21, 869-883 (1998).

*Cummings DE, Brandon EP, Planas JV, Motamed K, Idzerda RL, McKnight GS. Genetically lean mice result from targeted disruption of the *RIIβ* subunit of protein kinase A. Nature 382, 622-626 (1996).*

*Brandon EP, et al. Defective Motor Behavior and Neural Gene Expression in *RIIβ*-Protein Kinase A Mutant Mice. The Journal of Neuroscience 18, 3639-3649 (1998).*

*Yang Y, Takeuchi K, Rodenas-Ruano A, Takayasu Y, Bennett MVL, Zukin RS. Developmental switch in requirement for PKA *RIIβ* in NMDA-receptor-dependent synaptic plasticity at Schaffer collateral to CA1 pyramidal cell synapses. Neuropharmacology 56, 56-65 (2009).*

*Amieux PS, et al. Compensatory regulation of *RIα* protein levels in protein kinase A mutant mice. Journal of Biological Chemistry 272, 3993-3998 (1997).*

Yang L, McKnight GS. Hypothalamic PKA regulates leptin sensitivity and adiposity. Nature communications 6, 8237 (2015).

Minor Points:

[Comment 6] Line 87: The authors state that mainly *RIα*, *RIβ*, and *RIIβ* are highly expressed in the brain. This might apply to some neurons in the brain, but *RIIα* is certainly also expressed in many neuronal and non-neuronal cells in the brain.

[Answer 6] Thanks for the suggestion. We have answered in the *Comment 1 from reviewer No 1* and modified the statements in revised manuscript (see lines 88-89).

[Comment 7] Line 169: Please state if the used AAV drives expression from a neuronal promoter or globally in all cell types of the hippocampal DG area. The methods section states *CMV bGlobin-MCS-3FLAG-2A-.....*, which appears not neuron-specific.

[Answer 7] We have added the information that “we mutated the serine 112 to alanine in the *RIIβ* subunit (AAV-*RIIβ*-S112A, a P-site mutant that cannot be phosphorylated) with a broadly expressed promoter CMV, to impede the auto-phosphorylation in the DG area of the murine hippocampus.” (see lines 174-177)

[Comment 8] Line 179-182: The sentence structure should be revised.

[Answer 8] We have improved the English writing in the revised manuscript.

[Comment 9] Line 210: change “a significant less seizure progression” to “significantly less seizure progression”

[Answer 9] Thanks for the suggestion. We have revised it as reviewer's requirement.

[Comment 10] Line 250: *et al.*, reference is missing.

[Answer 10] We have added the reference.

[Comment 11] Line 260: Change "AAV-vector-tranducted" to "AAV-vector-transduced"

[Answer 11] We have revised it as reviewer's requirement.

[Comment 12] Line 346-350: *The sentence structure should be revised.*

[Answer 12] We have improved the English writing in the revised manuscript.

[Comment 13] Line 355: "dynamics" to "dynamic"

[Answer 13] We have revised it as reviewer's requirement.

[Comment 14] Figure 1, line 798: *Please indicate which tissue has been analyzed and explain abbreviations NC and EP.*

[Answer 14] We have revised it as reviewer's requirement.

[Comment 15] Figure 2, line 813: *A bit difficult to follow, maybe "Downregulation of RII β autophosphorylation in vivo leads to increased phosphorylation of CREB and PKA substrates as well as increased epileptic seizure susceptibility in mice."*

[Answer 15] We have revised it as reviewer's requirement.

[Comment 16] Figure 2H: *Indicate the applied statistic test in the figure legend.*

[Answer 16] We have revised it as reviewer's requirement.

Reviewer #2:

In this study, the authors addressed the mechanistic role of PKA-RII β subunit (one of the PKA regulatory subunits) in the genesis of epileptic seizures in both human patients and mice. PKA-RII β subunit autophosphorylation was decreased in both human and mouse samples from epileptic hippocampi, and RII β phosphorylation levels negatively correlated with the PKA catalytic activity. KA seizures were increased in mice infused with AAV-RII β -S112A (a persistent phosphorylation form of RII β), while RII β ko mice showed a lower susceptibility to KA seizures. AAV-RII β -S112A transduction in hippocampal neurons decreased miniature IPSCs (mIPSCs) and enhanced the neuronal intrinsic excitability, while increased mIPSCs and reduced neuronal intrinsic excitability was observed in hippocampal cells from RII β ko mice. Taken together, this data indicates that autophosphorylation of RII β subunit is a key player controlling hippocampal excitability.

This is an interesting paper, that likely deserves publication in Communications Biology. However, several parts of the manuscript need clarification and amendments.

Major issues

[Comment 1] page 6, lines 144-148. The following statement should be modified according to the available literature: "After a delay of a few weeks, known as the latent period (during which animals appear to be normal), spontaneous overt behavioral seizures occur (defined as the onset of chronic TLE). This model is widely used because many of the clinical and pathological features of the human disorder (including the latent period) can be reproduced". The systemic (intraperitoneal) KA model is known to induce a latent phase followed by chronic epilepsy (recurrent seizures) in the rat (to which the quoted studies refer) but not in the mouse. In addition, the short and long term effects of systemic KA markedly vary across mouse strains (see the classical studies by P-E. Schauwecker). Specifically, C57BL/6 mice are quite resistant to KA and do not develop chronic seizures when given 30mg/kg KA. So, as to my current knowledge, systemic KA cannot be considered a valid model of TLE.

[Answer 1] Thanks for the reviewer's comments. We agreed with the reviewer that one may have to struggle weeks before getting stable results especially with intraperitoneal injection of kainic acid because mice are less likely to develop spontaneous seizures compared to rats (Levesque, Avoli, & Bernard, 2016; McKhann, Wenzel, Robbins, Sosunov, & Schwartzkroin, 2003; Schauwecker, 2003; Schauwecker & Steward, 1997). Therefore, we have revised our statements in the manuscript. Actually, in our studies, we have only used the acute C57BL/6 mouse seizure model to evaluate the development and severity of KA-induced status epilepticus, not spontaneous epilepsy. The acute seizure model has been widely used for susceptibility study of epilepsy (Huang, Walker, & Shah, 2009; McKhann et al., 2003; Xiao et al., 2018). In addition, although the mouse KA model does not show stable latent period, there are several molecular and cellular changes in mouse brains following the status epilepticus such as axon sprouting, structural changes in pre- and postsynaptic receptors, changes in voltage-gated ion channels, alterations of homeostatic mechanisms and neuronal degeneration (Huang et al., 2009; Xiao et al., 2018). Despite resistant to cell death and synaptic found in TLE animal model (X. Zhang et al., 2002), considering that obvious neuronal death or mossy fiber sprouting may contribute to chronic seizures, it would be reasonable to test pathological changes during the latent period. Thanks again for the reviewer's suggestion, we have revised the statement about mouse TLE model.

Levesque, M., M. Avoli, and C. Bernard, *Animal models of temporal lobe epilepsy following systemic chemoconvulsant administration. J Neurosci Methods*, 2016. 260: p. 45-52.

McKhann, G.M., 2nd, et al., *Mouse strain differences in kainic acid sensitivity, seizure behavior, mortality, and hippocampal pathology. Neuroscience*, 2003. 122(2): p. 551-61.

Schauwecker, P.E. and O. Steward, *Genetic determinants of susceptibility to excitotoxic cell death: implications for gene targeting approaches. Proc Natl Acad Sci U S A*, 1997. 94(8): p. 4103-8.

Schauwecker, P.E., *Genetic basis of kainate-induced excitotoxicity in mice: phenotypic modulation of seizure-induced cell death. Epilepsy Res*, 2003. 55(3): p. 201-10.

Huang, Z., M.C. Walker, and M.M. Shah, *Loss of dendritic HCN1 subunits enhances cortical excitability and epileptogenesis. J Neurosci*, 2009. 29(35): p. 10979-88.

Xiao, K., et al., *ERG3 potassium channel-mediated suppression of neuronal intrinsic excitability and prevention of seizure generation in mice. J Physiol*, 2018. 596(19): p. 4729-4752.

Zhang, X., et al., *Relations between brain pathology and temporal lobe epilepsy. J Neurosci*, 2002. 22(14): p.

6052-61.

[**Comment 2**] *In line with the above mentioned issue, the genetic background of RII β ko mice (C57BL6? other?) should be clearly reported in the Methods section.*

[**Answer 2**] The genetic background of RII β KO mice were in a C57BL/6 background (Zheng et al., 2013). We have clarified the strain of RII β KO mice in the **Experimental procedures: Animals** section (see **lines 495-499**).

Zheng R, et al. Deficiency of the RIIbeta subunit of PKA affects locomotor activity and energy homeostasis in distinct neuronal populations. Proceedings of the National Academy of Sciences of the United States of America 110, E1631-1640 (2013).

[**Comment 3**] *page 8 lines 215-216. Again, the following statement should be amended: "These results suggested that PKA activity in epileptic foci may be causally linked to epileptogenesis". This is not correct; the current study, as designed, is not addressing epileptogenesis, but only the occurrence of acute seizures after KA administration in mice.*

[**Answer 3**] We have revised the statement as reviewer's suggestion.

[**Comment 4**] *The Discussion is a mere summary of the results, with minimal interpretation/speculation. Please modify and expand.*

[**Answer 4**] We have modified and expanded the discussion such as **lines 318-352 & 366-384**.

Minor issues

[**Comment 5**] *page 7, line 171. Please better define "in most region of DG": where different cell types transduced by AAV-RII β -S112A ?*

[**Answer 5**] We have revised this sentence as "We observed that AAV-RII β -S112A was robustly expressed in almost all cells in the DG area on the third week after the injection (Fig. 2A, Supplementary Fig. 2A)" (see **lines 177-178**).

[**Comment 6**] *Methods: please describe how the different hippocampal subfields (DG, CA1, CA3) were dissected for western blot.*

[**Answer 6**] Acute horizontal hippocampal slices containing dorsal hippocampus at 350- μ m thickness were cut and collected by using a vibratome (Leica VT1200S, Germany). Medial EC, CA1, CA3a-b, and DG subfields were rapidly microdissected and stored at -80°C until use based on previous studies (Grooms, Opitz, Bennett, & Zukin, 2000; Sultan, 2013). We dissected the specific subregions from each slice containing dorsal hippocampus as sketch map (from Grooms et al. PNAS, 2000, Fig.6 Inset in A (Grooms et al., 2000)) below. We have added the information (see the **Experimental procedures: Total protein extraction and western blot analysis** section, **lines 599-605**)

Sultan FA. Dissection of different areas from mouse hippocampus. *Bio-protocol* 3, (2013).

Grooms SY, Opitz T, Bennett MV, Zukin RS. Status epilepticus decreases glutamate receptor 2 mRNA and protein expression in hippocampal pyramidal cells before neuronal death. *Proceedings of the National Academy of Sciences of the United States of America* 97, 3631-3636 (2000).

Reviewer #3 (Remarks to the Author):

In this manuscript Zhang et al., analyze the involvement of the regulatory subunit RII-beta of the protein kinase A, in modulating the generation of seizures. PKA is an important kinase that it is extensively expressed across multiple brain regions and its function is very important to modulate critical physiological functions. PKA is formed by two regulatory subunits and two catalytic. Its activation is mediated for binding of cAMP to the regulatory subunit. The authors pay attention to the role of PKA-RII β subunit as a possible key factor in the regulation of neuronal excitability. They have found a reduction in the autophosphorylation of PKA-RII β , both in human and mouse model of temporal lobe epilepsy. This reduction correlates with an increase in PKA activity and CREB phosphorylation. Complementary results are obtained using different strategies, including the use of KO-PKA-RII β mice or in contrast, via the suppression of the subunit autophosphorylation. Finally, they observe how an increase in PKA activities enhance the neuronal intrinsic excitability in hippocampal DG granule cells and seizure susceptibility. In contrast, a reduction of PKA activities observed in a KO-PKA-RII β can perform a reduction in neuronal intrinsic excitability and less seizure susceptibility.

Overall this is a strong manuscript, but a few areas discussed below need to clarify or improve.

Major comments:

[Comment 1] *In the abstract authors mention that they induce an increase in PKA activities generating a persistent phosphorylation form of RII β using a transduction with adenovirus. In a deep reading of the manuscript is clear what the authors have made but it is confusing in a first view. A reader could think that it is permanent phosphorylation in the Serine 112 site that mention just above. It would necessary to clarify it in order to understand that it is an inactivation strategy via suppression of autophosphorylation.*

[Answer 1] We have revised it as reviewer's suggestion (see the **Abstract** section, **lines 44-45**).

[Comment 2] *RII- β is anchored to adenosine kinase anchoring proteins in the cell membrane. This way, RII β subunit regulates the phosphorylation level of neurotransmitter receptors and ion channels. There is a recent article (Tiwari et al, 2019. J.Neurosci) in which describes that the KCa-sAHP reduction in hippocampal neurons in TLE is due to the downregulation of KCa3.1 channels, mediated by PKA. Could you discard that some of the effects that you find in the excitability are not mediated by alterations in ion channels?*

[Answer 2] Thanks for reviewer's comments. We cannot rule out the effect which are mediated by other ion channels. As reviewer mentioned in the comments, Tiwari *et al.* (Tiwari, Mohan, Biala, & Yaari, 2019) found that in hippocampal CA1 neurons from TLE rats, PKA-mediated suppression of the slow afterhyperpolarizing (sAHP) was induced by downregulation of KCa3.1. In our study, we could also find the similar results that the enhanced PKA activities by reduced autophosphorylation of RII β *in vivo* could lead increased neuronal firings with an elevated sAHP (Fig. 5I and Supplementary Fig. 3B, statistical results not shown), while the decreased PKA activities in RII β null mice showed reduced neuronal firings and a trend for lower sAHP (Fig. 6I and Supplementary Fig. 4B, statistical results not shown). Thus, the sAHP-KCa3.1 current might be also mediated by the autophosphorylation of RII β , which is consistent with the previous study (Tiwari et al., 2019). We have discussed these results in discussion sections (see **lines 460-470**).

The AAV-RII β -S112A-transduced neurons have more depolarized RMP values, which is the main factor for increased neuronal excitability (Fig. 5), as these neurons fire the same number of APs as control neurons when they were artificially held at the membrane potential of -80 mV (Supplementary Fig. 3). Consistently, RII β ^{-/-} neurons have hyperpolarized RMP values and reduced neuronal intrinsic excitability (Fig. 6 and Supplementary Fig. 4). However, the differences found in AP firings and input insistence values as well as fAHP at RMP, remained only in RII β ^{-/-} rather than AAV-RII β -S112A-transduced group when cells were artificially held at the membrane potential of -80 mV (Supplementary Fig. 4), thus these changes may be due to the developmental changes for neuronal intrinsic properties (Pedarzani & Storm, 1993; Vogalis, Harvey, & Furness, 2003; Yasuda, Huang, & Tsumoto, 2008), either by alterations in ion channels by PKA subunits compensatory for RII β deficiency or non-PKA compensatory mechanism. It seems that the changes in AP amplitude, found either in AAV-RII β -S112A-transduced neurons only holding at RMP but not at -80 mV (Fig. 5L and Supplementary Fig. 3E) or in RII β ^{-/-} neurons only holding at -80 mV but not at RMP (Fig. 6L and Supplementary Fig. 4E), is thus unlikely mediated by alterations in ion channels due to direct changes in PKA-RII β . We also discuss these results in revised manuscript (see **lines 439-460**).

Tiwari MN, Mohan S, Biala Y, Yaari Y. Protein Kinase A-Mediated Suppression of the Slow Afterhyperpolarizing KCa3.1 Current in Temporal Lobe Epilepsy. *The Journal of Neuroscience* 39, 9914-9926 (2019).

Pedarzani P, Storm JF. Pka mediates the effects of monoamine transmitters on the K⁺ current underlying the slow spike frequency adaptation in hippocampal neurons. *Neuron* 11, 1023-1035 (1993).

Yasuda H, Huang Y, Tsumoto T. Regulation of excitability and plasticity by endocannabinoids and PKA in developing hippocampus. *105*, 3106-3111 (2008).

Vogalis F, Harvey JR, Furness JB. J. Physiol. PKA - mediated inhibition of a novel K⁺ channel underlies the slow after - hyperpolarization in enteric AH neurons. 548, 801-814 (2003)

[Comment 3] *The data obtained in the patient samples is really clear, but I am struck that the controls used are adjacent normal tissues. Have you had the opportunity to analyze the status of RII β in the hippocampus of control patients? Could there be differences between areas? In mice model comparison is made with wild types animals in the same areas, have you test the effect in adjacent areas?*

[Answer 3] In the mouse experiment of seizure model, we have not tested the expression of RII β in adjacent area. The reason is that in mouse TLE model, the seizures could be initiated in hippocampus, entorhinal cortex or both. It is difficult to distinguish the foci and adjacent normal tissues by implanting multiple recording electrodes into small mouse brain, thus the feasible way is to take the littermates administrated with saline (all other conditions should be the same as the experimental seizure model mice) as control mice, and the corresponding tissues as control tissues.

In addition, we do not have ethical approval, which allow us to obtain or analyze the hippocampal tissues from control patients. In compromise, before surgical resection of hippocampi, the multi-channels intracranial EEG electrodes were implanted to localize the origin of epilepsy. The “epileptic tissues” in our study were from originating sites of spontaneous seizure which were determined by their electrical seizure-like activities recorded with intracranial EEG electrodes. Tissues adjacent to “epileptic tissues” serve as the “controls”, which are normally the adjacent entorhinal cortical area. The control tissues could also show abnormal electrical activities during a seizure, but the occurrence of such activities is significantly later than that in epileptic tissues and we thus consider it as secondary effect.

[Comment 4] *In Fig.2, authors show as inactivation of autophosphorylation in serine 112 increases the activity of PKA, the rate of phosphorylated CREB protein, the p-PKA substrates, and the seizures. However, there is missing of the manuscript a discussion or mention of the results found using the kainate mice model, concretely the great reduction in p-RII levels and in consequence in the PKA activity (comparison between control and epi of S112A). An explanation of these results would be appropriate to include. Do you think that there are other mechanisms independent of RII-beta? Are these results a consequence of the method used?*

[Answer 4] This is very good question, which is similar to the question raised by *reviewer No1*. To explain the mechanism underlying the increased PKA activity in TLE patients and in KA-induced mouse seizure model, we drew a cartoon as follow to illustrate the possible pathological processes under TLE. In details, the cartoon shows that in physiological condition the PKA-C subunits of can autophosphorylate RII β subunits using MgATP, and the phosphorylated RII β can turn back into unphosphorylated RII β using protein phosphatases (PPs). The PKA holoenzyme exists in a phosphorylated state due to the rapidly autophosphorylation. When intracellular levels of cAMP rise, the phosphorylated and de-phosphorylated RII β homodimer start to associate with cAMP and thus releases the active C subunits and increase the PKA activity. Conversely, when cAMP levels fall, the phosphorylated and de-phosphorylated RII β can

reassociate with C to reconstitute into an inactive tetrameric holoenzyme (Rangel-Aldao & Rosen, 1976; Ping Zhang et al., 2015; P. Zhang et al., 2015; P. Zhang et al., 2012). It was reported that dephosphorylated RII β can associate with C subunit at least 5 times more rapidly compared with the association rate of phosphorylated RII β with C subunit (Rangel-Aldao & Rosen, 1976). This means the dephosphorylated RII β subunits can associate with C subunit more efficiently, when compared with phosphorylated RII β . Alternatively, the dephosphorylated RII β can also be rapidly dephosphorylated by PPs, thus, greatly facilitating its capacity to reassociate with C and generate the inactive holoenzyme. Therefore, the two detectable forms by west blot analysis are composed of “p-RII β ” mainly derived from phosphoholoenzyme, and the disassociated “RII β ” (subtracted by p-RII β from total RII β), which in theory approximately equals to the amount of the disassociated C subunits, in positive proportion to the PKA activity. However, under TLE condition, the abnormal neuronal activities might increase cAMP levels, reduce the RII β subunit’s autophosphorylation or increase the PPs activity, and hence decrease the proportion of phosphoholoenzyme (detected as “p-RII β ” in western blot analysis), thus to destabilize the holoenzyme, resulting in more disassociated C subunits, which is reflected by the disassociated RII β subunits. The results suggested that higher proportion of RII β subunit staying in the dephosphorylated states, correspond to higher PKA activity state. Here, we also discussed the possible mechanism in revised manuscript (see lines 326-352).

In addition, in this study, we found that the decreased autophosphorylation of RII β subunit of PKA and increased PKA activity is causally linked to the acute generation of temporal lobe seizures, although there may be other mechanisms independent of RII β . Taken together, these findings are not the consequence of the method used, on the contrary, we suggest that the increased PKA activities may underlie the generation and development of TLE.

Rangel-Aldao, R. and O.M. Rosen, Dissociation and reassociation of the phosphorylated and nonphosphorylated forms of adenosine 3':5' -monophosphate-dependent protein kinase from bovine cardiac muscle. *Journal of Biological Chemistry*, 1976. 251(11): p. 3375-80.

Zhang, P., et al., Structure and allostery of the PKA RIIbeta tetrameric holoenzyme. *Science*, 2012. 335(6069): p. 712-6.

Zhang, P., et al., Discovery of Allostery in PKA Signaling. *Biophys Rev*, 2015. 7(2): p. 227-238.

Zhang, P., et al., Single Turnover Autophosphorylation Cycle of the PKA RII β Holoenzyme. *PLOS Biology*, 2015. 13(7): p. e1002192.

[Comment 5] KO-RII β mice have a decreased level of p-CREB and p-substrates of PKA. The R subunits act as an intrinsic inhibitor of the catalytic subunits but also protect it to degradation. In

previous reports, it has described as RII beta deficiency leads to decreased catalytic subunits and PKA activity. I would recommend you include this data in the discussion in view of your results.

[Answer 5] By analyzing of a variety of mammalian tissues, it has revealed significant differences in the ratio of type I (RI-containing) to type II (RII-containing) holoenzyme. In mice, brain and adipose tissues contain principally the type II holoenzyme (J. Corbin & Keely, 1977; J. D. Corbin, Keely, & Park, 1975; J. D. Corbin, Sugden, Lincoln, & Keely, 1977). In adipocytes, the RII β subunits preferentially associate with C, leaving a pool of free RI α that is rapidly degraded. Type I holoenzyme is only formed when the level of C subunits exceeds the level of RII subunits (in this case caused by the loss of RII β). In addition, global knockout RII β in mice is associated with a compensatory increase of the RI isoform in various tissues, such as hypothalamus, brown adipose tissue, white adipose tissue, etc (Amieux et al., 1997) (L. Yang & McKnight, 2015). In this situation, RI α can successfully compete for binding to the pool of free C subunits and is therefore stabilized in a holoenzyme complex. However, the increases in these R subunits did not compensate fully for the loss of RII β , for example, there is a 30% loss of R subunits overall in mutant white adipose tissue, as assessed by total cAMP-binding capacity (Planas, Cummings, Idzerda, & McKnight, 1999), leaving the unbound C subunits susceptible to proteolysis. Thus, the catalytic subunits of PKA (C α and C β) were reduced dramatically, as predicted by the decreased level of p-CREB and p-substrates of PKA. We have discussed these results in revised manuscript (see lines 429-444).

Corbin JD, Keely S, Park CR. The distribution and dissociation of cyclic adenosine 3':

5'-monophosphate-dependent protein kinases in adipose, cardiac, and other tissues. Journal of Biological Chemistry 250, 218-225 (1975).

Corbin JD, Sugden P, Lincoln TM, Keely S. Compartmentalization of adenosine 3': 5'-monophosphate and adenosine 3': 5'-monophosphate-dependent protein kinase in heart tissue. Journal of Biological Chemistry 252, 3854-3861 (1977).

Corbin J, Keely S. Characterization and regulation of heart adenosine 3': 5'-monophosphate-dependent protein kinase isozymes. Journal of Biological Chemistry 252, 910-918 (1977).

Amieux PS, et al. Compensatory regulation of RI α protein levels in protein kinase A mutant mice. Journal of Biological Chemistry 272, 3993-3998 (1997).

Yang L, McKnight GS. Hypothalamic PKA regulates leptin sensitivity and adiposity. Nature communications 6, 8237 (2015).

Planas JV, Cummings DE, Idzerda RL, McKnight GS. Mutation of the RII β subunit of protein kinase A differentially affects lipolysis but not gene induction in white adipose tissue. Journal of Biological Chemistry 274, 36281-36287 (1999).

Minor comments:

[Comment 6] *Figure legends. I have noted that some information is missing, for example, the explanation of the acronyms used. For example, fig 1A and C, is not described as the meaning of NC or EP. Please review the legend to complete the information.*

[Answer 6] We have revised it as reviewer's suggestion.

Reviewers' comments:

Reviewer #1 (Remarks to the Author):

[To Answer 1]

The used anti-phospho-RII β antibody from BD is clearly not specific for RII β . This is shown in Figure 3A of the mentioned publication (see Isensee et al., J Cell Sci, 2014). As I stated already, the inhibitory sites of RII subunits are well preserved in both RII isoforms and it is likely not possible to discriminate between pRII α and pRII β . If the authors state that the pRII β antibody is specific, they could prove this by absence of a pRII β signal in hippocampal lysates of RII β knockout mice (which they have access to). This would have been a very simple and fast experiment.

Concerning the expression pattern of RII subunits, I fully agree that the work Cadd & McKnight is pioneering great work, but newer RNA-Seq datasets for instance may have also been used by the authors to support the claimed expression pattern of RII subunits.

[To Answer 2]

Thank you for the helpful cartoon and extended discussion. I think it is clear for the reader that the interpretation is not simple.

[To Answer 3]

The added results support that RII β is not expressed in GFAP-positive astrocytes. Transduction with the AAV-RII β -S112A, however, could lead to expression of RII β in these cells, because an unspecific CMV promoter was used.

[To Answer 4]

Thank you for adding the blots to the supplement. Unfortunately, my major question was not addressed. It is not clear for the reader, why the pRII blot shows additional bands only in Ctrl s transduced with the empty AAV-Vector. It also not clear why additional bands appear on the RII β blot only in the experimental seizure model. This is quite confusing, because the reader expects that transduction with the AAV-RII β -S112A leads to higher RII β levels. In these samples, however, the RII β bands are unchanged. Are the additional bands splicing variants of RII β that are specifically induced in the experimental seizure model?

Reviewer #2 (Remarks to the Author):

I am very satisfied with the revisions introduced by the authors, and I now consider the paper acceptable for publication. If compatible with the journal format, I would suggest to include Suppl Fig. 5 (illustrating the proposed mechanism underlying the increased PKA activity in TLE) as the last Figure of the main body of the manuscript and not as a supplementary one.

Reviewer #3 (Remarks to the Author):

The authors have satisfactorily answered all my questions. I think that after the review, the authors have significantly improved the quality of the article. It is worth noting the discussion, since

important aspects have been included now that they were not included in the previous version. Furthermore, the included drawings to explain the possible pathological process in TLE are very useful.

Point-to-point response to reviewers' comments

We would like to thank the reviewers for taking the time to consider our manuscript carefully and thoughtfully. We appreciate the comments and have done necessary experiments to address these concerns, which we feel have substantially improved and strengthened our manuscript. Specific comments are addressed below:

Reviewer #1 (Remarks to the Author):

Specific Comments:

[Comment 1] *The used anti-phospho-RII β antibody from BD is clearly not specific for RII β . This is shown in Figure 3A of the mentioned publication (see Isensee et al., J Cell Sci, 2014). As I stated already, the inhibitory sites of RII subunits are well preserved in both RII isoforms and it is likely not possible to discriminate between pRII α and pRII β . If the authors state that the pRII β antibody is specific, they could prove this by absence of a pRII β signal in hippocampal lysates of RII β knockout mice (which they have access to). This would have been a very simple and fast experiment. Concerning the expression pattern of RII subunits, I fully agree that the work Cadd & McKnight is pioneering great work, but newer RNA-Seq datasets for instance may have also been used by the authors to support the claimed expression pattern of RII subunits.*

[Answer 1] Thank you very much for pointing it out. We fully agree with you that the inhibitory sites of RII subunits are well preserved in both RII isoforms and it is likely impossible to discriminate between p-RII α and p-RII β . However, as we have stated that RII β is the predominant subunit in hippocampus, while RII α were found almost exclusively in the media habenula nuclei (Cadd & McKnight, 1989). Therefore, we may use this antibody to discriminate between p-RII α and p-RII β safely in hippocampal lysates.

We really respect your opinion, of course. Following your rigorous comments, we performed new experiments to reconfirm the specificity of the p-RII β antibody. Here, we would like to present our new western blot results (as shown below, also see **Supplementary Fig. 2F**). The results we obtained demonstrated that the RII β and p-RII β signals were absent from the hippocampal lysates of RII β knockout mice, indicating the specificity of the p-RII β antibody in the hippocampal tissues.

Gratefully, we fully agreed with you that RNA-Seq datasets is a great technique for studying the expression pattern of RII subunits. In fact, we were also interested in this question. Therefore, in our future work, we would like to apply RNA-Seq and single-cell sequencing to directly

address the expression pattern of RII subunits of all cell types in the hippocampus. However, combined with our new results, we think it is sufficient to characterize the expression pattern of RII subunits in hippocampal tissues.

Cadd G, McKnight GS. Distinct patterns of cAMP-dependent protein kinase gene expression in mouse brain. Neuron 3, 71-79 (1989).

[Comment 2] Thank you for the helpful cartoon and extended discussion. I think it is clear for the reader that the interpretation is not simple.

[Answer 2] Thank you very much for your kind words.

[Comment 3] The added results support that RII β is not expressed in GFAP-positive astrocytes. Transduction with the AAV-RII β -S112A, however, could lead to expression of RII β in these cells, because an unspecific CMV promoter was used.

[Answer 3] Thank you for your comments, and we totally agree with your point that the transduction with the AAV-RII β -S112A may lead to expression of RII β in astrocytes due to the unspecific CMV promoter. However, as we presented in the manuscript, the transduced florescence by AAV9-RII β -S112A was mainly localized in the cells especially within the dentate gyrus granule cell layer (as shown below, also see **Supplementary Fig. 2C**). Our result is consistent with a previous study (Klein, Dayton, Tatom, Henderson, & Henning, 2008), which demonstrated a relative specificity for the transduction pattern of the CMV promoter via AAV9 vector by I) AAV9-CMV transduction incorporating cells with neuronal morphology in the hippocampal subregions among different AAV vectors, II) and overall strength of the florescence signal on sections matching the western and biophotonic data.

Considering the absence of endogenous RII β in astrocytes, our opposite results obtained from RII β ^{-/-} mice were thus mainly from neurons rather than astrocytes. Combining all these data, we did not further apply the neuronal-specific promoter to control the expression of AAV-RII β -S112A.

Klein, R. L., Dayton, R. D., Tatom, J. B., Henderson, K. M., & Henning, P. P. (2008). AAV8, 9, Rh10, Rh43 Vector Gene Transfer in the Rat Brain: Effects of Serotype, Promoter and Purification Method. Molecular Therapy, 16(1), 89-96. doi: <https://doi.org/10.1038/sj.mt.6300331>

[Comment 4] Thank you for adding the blots to the supplement. Unfortunately, my major question was not addressed. It is not clear for the reader, why the pRII blot shows additional bands only in Ctrl's transduced with the empty AAV-Vector. It also not clear why additional bands appear on the RII β blot only in the experimental seizure model. This is quite confusing, because the reader expects that transduction with the AAV-RII β -S112A leads to higher RII β levels. In these samples,

however, the RII β bands are unchanged. Are the additional bands splicing variants of RII β that are specifically induced in the experimental seizure model?

[Answer 4] Thank you very much for your valuable critics. As you mentioned, the p-RII blot showed additional bands only in Ctrl transduced with the empty AAV-Vector, and the total RII β additional bands appeared only in the experimental seizure model. Based on the molecular weight of the RII β protein (53 kDa), we only calculated the bands between 40 kDa and 55 kDa, and thus the transduction with the AAV-RII β -S112A did not lead to higher RII β levels. In fact, our current experimental results cannot provide a perfect explanation, and importantly, there is currently no relevant information available for our reference about the additional bands. Given that the expression levels of the PKA-RI and PKA-C subunits are not changed by AAV-RII β -S112A transduction, we speculated that the transduced RII β -S112A could inhibit the endogenous RII β expression to maintain the balance of the functional RII β subunit. Gratefully, we were also highly agreed with you that the additional bands might be splicing variants of RII β and seemed to be specifically induced in the experimental seizure model. Following your comments, we would like to explain the phenomenon in our future work. Nevertheless, the additional bands did not affect the conclusion that PKA-RII β autophosphorylation modulates the PKA activities and regulates the generation of mouse seizures.

Reviewer #2 (Remarks to the Author):

[Comment 1] *I am very satisfied with the revisions introduced by the authors, and I now consider the paper acceptable for publication. If compatible with the journal format, I would suggest to include Suppl Fig. 5 (illustrating the proposed mechanism underlying the increased PKA activity in TLE) as the last Figure of the main body of the manuscript and not as a supplementary one.*

[Answer 1] Thanks for your thoughtful review of our work and kind words. Following your comments and the journal format, the manuscript has been accordingly modified (see **Fig. 7**).

Reviewer #3 (Remarks to the Author):

[Comment 1] *The authors have satisfactorily answered all my questions. I think that after the review, the authors have significantly improved the quality of the article. It is worth noting the discussion, since important aspects have been included now that they were not included in the previous version. Furthermore, the included drawings to explain the possible pathological process in TLE are very useful.*

[Answer 1] We appreciate your constructive and insightful comments, which greatly improved our manuscript.

Cadd, G., & McKnight, G. S. (1989). Distinct patterns of cAMP-dependent protein kinase gene expression in mouse brain. *Neuron*, 3(1), 71-79.

Klein, R. L., Dayton, R. D., Tatom, J. B., Henderson, K. M., & Henning, P. P. (2008). AAV8, 9, Rh10, Rh43 Vector Gene Transfer in the Rat Brain: Effects of Serotype, Promoter and Purification Method. *Molecular Therapy*, 16(1), 89-96. doi:<https://doi.org/10.1038/sj.mt.6300331>